# A three-level regulatory mechanism of the aldo-keto reductase subfamily AKR12D

Zhihong Xiao[1], Jinyin Zha[2], Xu Yang [®][1], Tingting Huang [®][1], Shuxin Huang[1], Qi Liu[1], Xiaozheng Wang[1], Jie Zhong[2], Jianting Zheng [®][1], Rubing Liang[1], Zixin Deng [®][1], Jian Zhang [®][2] ✉, Shuangjun Lin [®][1,3,4] ✉ & Shaobo Dai [®][1] ✉

Modulation of protein function through allosteric regulation is central in biology, but biomacromolecular systems involving multiple subunits and ligands may exhibit complex regulatory mechanisms at different levels, which remain poorly understood. Here, we discover an aldo-keto reductase termed AKRtyl and present its three-level regulatory mechanism. Specifically, by combining steady-state and transient kinetics, X-ray crystallography and molecular dynamics simulation, we demonstrate that AKRtyl exhibits a positive synergy mediated by an unusual Monod-Wyman-Changeux (MWC) paradigm of allosteric regulation at low concentrations of the cofactor NADPH, but an inhibitory effect at high concentrations is observed. While the substrate tylosin binds at a remote allosteric site with positive cooperativity. We further reveal that these regulatory mechanisms are conserved in AKR12D subfamily, and that substrate cooperativity is common in AKRs across three kingdoms of life. This work provides an intriguing example for understanding complex allosteric regulatory networks.

Proteins are tunable and their activities are usually modulated by the perturbation of environmental signals[1,2]. The binding of effectors at one site transmits information spatially to remote sites through communication pathways, which is termed allostery[3–5]. Allostery is considered a general phenomenon with essential roles in biology[6–10], serving as a source of new drug targets[11,12] and protein design[13,14]. Despite its importance, allosteric mechanisms remain poorly understood, especially how allosteric modulation with multiple modulators and varying input strengths in enzyme catalysis is still enigmatic[15,16]. The classical MWC and Pauling-KNF models only cope with one allosteric modulator and the direction of the switch is unimodal[17,18], while enzyme catalysis typically involves more than one substrate and may exhibit more complex regulatory mechanisms[19–21]. Therefore, deep understanding of enzyme catalysis with multiple levels of allosteric regulation will further deepen our knowledge of how complex enzyme

systems respond to multiple environmental stimuli and their physiological significance.

Aldo-keto reductases (AKRs) are a superfamily of oxidoreductases capable of reducing carbonyl to primary and secondary alcohols, found in all phyla[22]. All characterized AKRs employ the canonical TIM barrel with conserved NADPH binding site[23–26] which undergoes conformational change upon cofactor binding, indicating potential allosteric regulation[27,28]. Cofactor binding in turn reshapes the substrate pocket that is promiscuous in most AKRs, and facilitates catalysis[29]. This feature is reminiscent of the fact that enzyme promiscuity may be one of the driving forces for the evolution of substrate-dependent allosteric regulation[30]. Moreover, in the presence or absence of NADP⁺, ligand (inhibitor) binding displayed reverse effects on the thermostabilities of AKR1A1 and AKR1B10, which are members of subfamily 1A and subfamily 1B of

[1]State Key Laboratory of Microbial Metabolism, Joint International Research Laboratory on Metabolic & Developmental Sciences, School of Life Sciences & Biotechnology, Shanghai Jiao Tong University, 800 Dongchuan Road, Shanghai 200240, China. [2]Medicinal Chemistry and Bioinformatics Center, Shanghai Jiao Tong University School of Medicine, Shanghai 200025, China. [3]Haihe Laboratory of Synthetic Biology, Tianjin 300308, China. [4]Frontiers Science Center for Transformative Molecules, Shanghai Jiao Tong University, Shanghai 200240, China. ✉e-mail: jian.zhang@sjtu.edu.cn; linsj@sjtu.edu.cn; sdai@sjtu.edu.cn

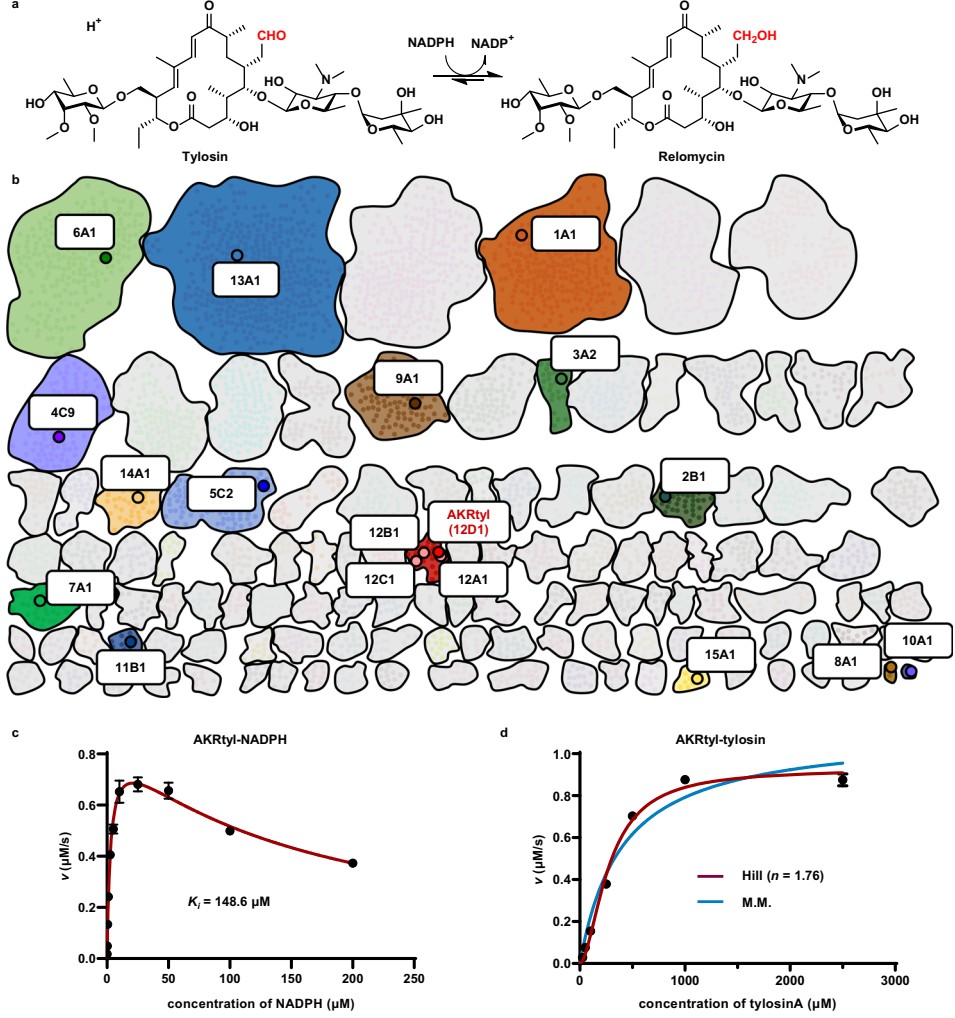

**Fig. 1 | AKRtyl functions in tylosin reduction and clusters in family 12 of AKRs.** **a** AKRtyl-catalyzed conversion of tylosin to relomycin through reduction of the C20 aldehyde group, using NADPH as a cofactor. **b** Abbreviated SSN of the AKR superfamily, highlighting selected sequence clusters and nodes. The full network (Supplementary Fig. 2) was generated from approximately 330,000 annotated AKRs (25000 UniRef50 cluster ID sequences) with an alignment score of 88. Each node represents one UniRef50 cluster. Sequence clusters and nodes are colored according to AKR families. Nodes containing functionally annotated sequences and representing each family are enlarged and colored. AKRtyl is colored in red and clustered with AKR12A1, AKR12B1, AKR12C1. **c** Steady-state kinetics of AKRtyl to cofactor NADPH reveals its inhibitory effect. Error bars indicate means ± SD ($n = 3$ independent experiments). **d** Steady-state kinetics of AKRtyl towards the substrate tylosin is well fitted by the Hill equation (red line; $R^2 = 0.9913$) but poorly fitted by the Michaelis-Menten equation (blue line; $R^2 = 0.9614$). The Hill coefficient ($n$) is 1.76, greater than 1. Error bars indicate means ± SD ($n = 3$ independent experiments).

the AKR superfamily and play important roles in aldehydes and ketones metabolism in humans[31,32]. These distinct effects of ligand binding were suggested to be allosteric regulation[33], but more evidence, including structural data, is lacking.

Tylosin is a macrolide antibiotic widely used as veterinary medicine and feed additive[34–36]. Reduction of the C20 aldehyde to primary alcohols generates the byproduct relomycin, which severely affects its activity and downstream derived products[37,38]. Although an early study thirty years ago reported a plausible NADPH-dependent reductase purified directly from tylosin-producing strain, its authentic sequence and phylogenetic origin, mode of action and potential regulatory mechanisms are still elusive[39].

In this work, we identify the enzyme AKRtyl, which is responsible for tylosin reduction. Phylogenetic analysis indicates that AKRtyl is the founding member of subfamily D within family 12 in the AKR superfamily. And we reveal an intriguing three-level regulatory mechanism regarding substrate tylosin and cofactor NADPH. This work will have exciting implications of our general understanding of enzyme catalysis and allosteric regulation.

## Results

### Assignment of tylosin reductase to subfamily AKR12D by phylogenetic analysis

Based on early reported two segment amino acid sequences of tylosin reductase[39], we obtained the complete protein sequence of tylosin reductase through NCBI database. Sequence analysis showed that tylosin reductase belonged to the AKR superfamily, so it can be named AKRtyl. To further specify its classification, we performed multiple sequence alignment and phylogenetic tree analysis of the annotated members from 15 families in the AKR superfamily. It showed that AKRtyl belonged to family 12 as it clusters with NDP-hexose-2,3-enoyl-reductase TylCII (AKR12A1), mycarose/desosamine reductase EryBII (AKR12B1) and dTDP-4-keto-6-deoxy-L-hexose-2,3-reductase AveBVII (AKR12C1) which are all from family 12 (Supplementary Fig. 1). A comprehensive sequence similarity network (SSN)[40] analysis of more than 330,000 AKRs showed the same result (Fig. 1b and Supplementary Fig. 2), that is, AKRtyl and the above three AKRs cluster together. AKRtyl has 53%, 49%, and 52% sequence identities with AKR12A1, AKR12B1, and AKR12C1, respectively, all below 60%. Therefore, AKRtyl

**Table 1 | Kinetic properties of AKRtyl and its mutants to cofactor NADPH and substrate tylosin.[a]**

| Ligands | Proteins | $K_m$ or $K_{0.5}$ (µM)[d] | $k_{cat}$ (s⁻¹)[d] | $k_{cat}/K_m$ (s⁻¹ µM⁻¹)[d] | $K_i$ (µM)[d] | Hill coeff $n$[e] |
|---|---|---|---|---|---|---|
| NADPH[b] | WT | 2.98 ± 0.16 | 4.38 ± 0.09 | 1470.0 × 10⁻³ | 148.6 ± 8.9 (Inhibition) | - |
| | W331A | 11.03 ± 1.09 | 0.18 ± 0.01 | 16.3 × 10⁻³ | 1541.0 ± 699.1 (Loss inhibition) | - |
| | W331A[f] | 8.33 ± 0.60 | 0.16 ± 0.01 | 19.3 × 10⁻³ | - | 1.17 ± 0.08 (Slight pos. coop.) |
| Tylosin[c] | WT | 214.51 ± 17.17 | 1.44 ± 0.05 | 6.71 × 10⁻³ | - | 1.76 ± 0.19 (Pos. coop.) |
| | WT[g] | 278.00 ± 12.32 | 4.65 ± 0.09 | 16.72 × 10⁻³ | - | 1.76 ± 0.11 (Pos. coop.) |
| | E193A | 501.70 ± 176.30 | 1.10 ± 0.15 | 2.20 × 10⁻³ | - | 0.99 ± 0.18 (Loss of coop.) |
| | E193W | 1147.98 ± 334.21 | 0.56 ± 0.07 | 0.49 × 10⁻³ | - | 1.05 ± 0.12 (Loss of coop.) |
| | R195A | 333.70 ± 32.77 | 1.28 ± 0.05 | 3.84 × 10⁻³ | - | 1.17 ± 0.09 (Reduced coop.) |
| | R195W | 690.83 ± 112.26 | 1.37 ± 0.10 | 1.98 × 10⁻³ | - | 1.08 ± 0.09 (Loss of coop.) |
| | Q254W | 329.70 ± 46.26 | 1.20 ± 0.65 | 3.63 × 10⁻³ | - | 1.12 ± 0.11 (Reduced coop.) |
| | R257A | 382.19 ± 57.17 | 1.39 ± 0.09 | 3.64 × 10⁻³ | - | 1.38 ± 0.20 (Reduced coop.) |
| | R257W | 372.01 ± 99.22 | 1.31 ± 0.13 | 3.52 × 10⁻³ | - | 0.95 ± 0.13 (Loss of coop.) |

All measurements were carried out in triplicate and are shown as mean ± SD (standard deviation, $n = 3$ independent experiments).
[a]This table lists kinetic data of the important mutants and the comprehensive mutants can be found in the Supplementary Table 2.
[b]The saturation concentration of tylosin in NADPH kinetic assays: 5 mM.
[c]The saturation concentration of NADPH in most tylosin kinetic assays: 200 µM.
[d]The Michaelis-Menten constant $K_m$ and $K_{0.5}$ in the Hill equation, the catalytic rate constant $k_{cat}$, the catalytic efficiency $k_{cat}/K_m$ and the inhibition constant $K_i$.
[e]The Hill coefficient $n$ of substrate binding as a measure of cooperativity ($n > 1$, positive cooperativity; $n = 1$, no cooperativity; $n < 1$, negative cooperativity).
[f]Kinetic data of the W331A mutant on NADPH are fitted with the Hill equation.
[g]The saturation concentration of NADPH in this tylosin kinetic assays: 25 µM.

does not belong to any of the existing subfamily so we named the subfamily AKR12D according to the classification nomenclature and systematically AKRtyl can be named AKR12D1[22].

## Distinct kinetic properties of AKRtyl towards cofactor and substrate

To confirm the function of AKRtyl, we characterized AKRtyl in vivo and in vitro. We first investigated the effect of AKRtyl on the production of the byproduct relomycin by analyzing the fermentation products of the *S. fradiae* TL-01 Δ*AKRtyl* strain. Compared to the wild-type strain, the ratio of relomycin to tylosin in the fermentation product decreased from 37.5% to 5.9%, suggesting that AKRtyl plays the role of reducing tylosin in vivo (Supplementary Fig. 4a) and is responsible for the byproduct relomycin production. Subsequently, we set out to characterize AKRtyl in vitro, N-terminally His₆-tagged AKRtyl was overexpressed in *Escherichia coli* BL21 and purified to homogeneity (Supplementary Fig. 5). In vitro biochemical assay was carried out with tylosin and NADPH, and the results showed that AKRtyl catalyzes the complete conversion of tylosin into relomycin (Supplementary Fig. 4b) which was verified by high-resolution mass spectrometry for exact molecular weight (Supplementary Fig. 4c, d). This result further confirms that AKRtyl is indeed capable of reducing tylosin (Fig. 1a).

We then performed detailed kinetic studies of AKRtyl for cofactor NADPH and substrate tylosin. All AKRs catalyze a sequential ordered bi–bi reaction in which cofactor binds first followed by tethering of substrate[22,41]. Surprisingly, in the presence of saturating tylosin, the kinetic curve of NADPH did not conform to the classical Michaelis-Menten equation but instead fitted well with the substrate inhibition equation with a $K_m$ value of 2.98 µM and a $K_i$ value of 148.6 µM (Table 1, Fig. 1c). This means that the high NADPH concentration will inhibit the enzyme, and this concentration has physiological significance, as the intracellular NADPH concentration can reach the sub-millimolar level[42,43]. Conversely, for substrate tylosin, AKRtyl displayed sigmoidal kinetics with a $K_{0.5}$ value of 278.00 µM and a Hill coefficient ($n$) of 1.76, indicating positive cooperativity (Table 1, Fig. 1d and Supplementary Fig. 7), and non-His-tagged form of AKRtyl still exhibits positive cooperativity (Supplementary Fig. 8 and Supplementary Table 3). This suggests that AKRtyl exhibits divergent properties in response to cofactor NADPH and substrate tylosin - binding of tylosin promoting the reaction, while NADPH is the opposite. Since many AKRs can reduce a wide range of aldehydes and ketones[22], we also

assessed the substrate scope of AKRtyl which showed that AKRtyl targets a broad-spectrum aldehydes including glyceraldehyde 3-phosphate and erythrose 4-phosphate which are important intermediates in glycolysis and the pentose phosphate pathway (Supplementary Table 1). Intriguingly, kinetic properties of some substrates were similar to those for tylosin with Hill coefficients ($n$) greater than 1, implying that the positive cooperativity of AKRtyl is a general mechanism irrespective of substrates (Supplementary Table 4 and Supplementary Fig. 7).

## Cofactor NADPH binding involves regulatory mechanism at two levels

To interrogate the molecular basis for different kinetic properties, we have successfully determined a series of high-resolution crystal structures of AKRtyl in different liganded state (Supplementary Table 6). These structures show that: (1) the monomer of AKRtyl exhibits the classic $(\alpha/\beta)_8$ fold (TIM barrel, derived from the structure of triose phosphate isomerase[44,45]) (Supplementary Fig. 10a–c); (2) the quaternary structure adopts an octameric architecture (consistent with size-exclusion chromatography coupled with multi-angle light scattering (SEC-MALS) experiment, Supplementary Fig. 6), consisting of two well-packed homotetramers face to face, with a 30° offset (Fig. 2a); (3) the C-terminal tail loops behave as hooks for anchoring by insertion into the active site of other subunits (Supplementary Fig. 10e, f).

We solved two AKRtyl crystal structures in the purified state. They differ in space group and asymmetric unit with resolution of 2.30 and 2.32 Å, respectively (Supplementary Table 6). Each structure contains cofactor NADP(H) carried from *E. coli* purification and we confirmed that the cofactor is in the oxidized state (Supplementary Fig. 9). We denote all cofactors below as NADP(H) for simplicity. The first structure contains a complete octamer in the asymmetric unit in which two subunits are bound with NADP(H) and belong to different tetramers (Supplementary Fig. 13e). Of the remaining six NADP(H) free subunits, a long cofactor-binding loop (approximately residues 230–241) are not fully observed in five subunits in the electron density map due to disorder, whereas this region is visible only in one subunit that presents an open state. In addition, a large cleft suitable for NADPH binding was formed above the TIM barrel in all these cofactor-free subunits. Cofactors are indeed bound in this cleft as demonstrated in the two subunits that contain NADP(H). NADP(H) adopts an extended

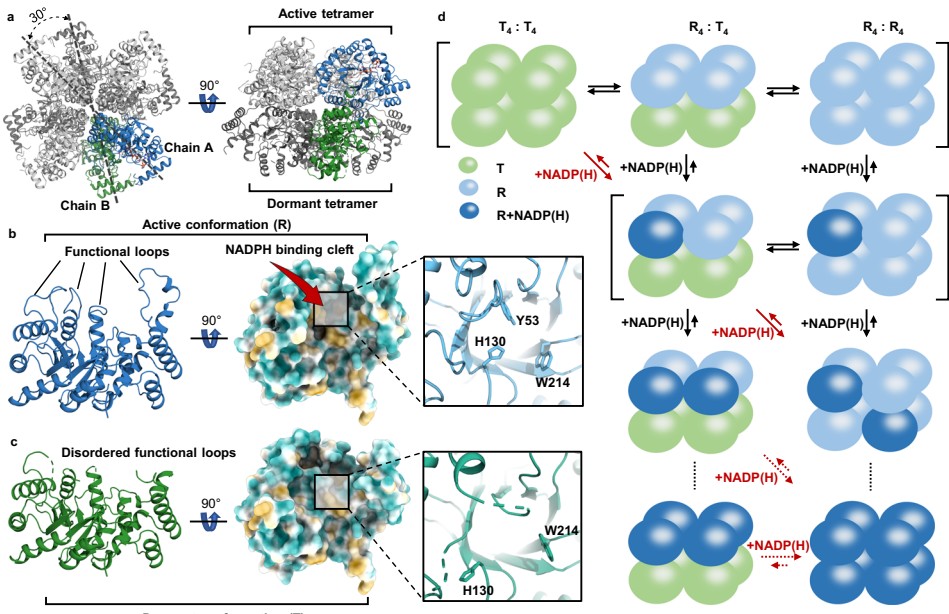

**Fig. 2 | AKRtyl binds NADP(H) employed a MWC paradigm allosteric regulation. a** Top and side views of the second purified state AKRtyl octamer that consists of two homotetramers enclasped face-to-face at 30° offset. One AKRtyl monomer of the two tetramers with different conformations (active: light gray, dormant: dark gray) is highlighted in blue (active) and green (dormant), respectively. **b, c** Cartoon (left), hydrophobic surface (middle) and the enlarged section of the active site (right) representations of the AKRtyl active monomer (R-state) and dormant (T-state) monomer. The R monomer with ordered functional loops and the cofactor binding pocket is clear but the T monomer with functional loops disorder and the cofactor binding pocket is blurred. **d** MWC allosteric model for the octameric AKRtyl. The allosteric oligomeric basic unit is tetramer and the

tetramer can populate only two conformational states in equilibrium: all subunits in R conformation or all subunits in T conformation. The upper square brackets indicate the conformational equilibrium of the two tetramers, corresponding to the three states of the octamer: $T_4T_4$, $R_4T_4$, and $R_4R_4$. The square brackets in the second row indicate that one tetramer in the octamer is conformationally stabilized to $R_4$ upon binding of NADP(H), and the other tetramer is in conformational equilibrium. The T-conformation subunit is shown as a green blob, the R-conformation subunit as a blue blob and the R-conformation subunit with bound NADP(H) as a dark blue blob. Red arrows indicate NADP(H)-driven equilibrium towards the R conformation. Dashed lines and dotted arrows indicate multiple processes.

conformation as it does in other AKRs[23-25] and the long cofactor binding loop in these two subunits are clearly ordered and showed a closed state with an 18 Å conformational change, which wrapped NADP(H) tightly like "safety belts" (Supplementary Fig. 10g, h). The nicotinamide ring on NADP(H) is π-π stacked against Trp214, and the carboxamide group makes contact with Ser160, Asn161, Gln186, and Trp331 which comes from neighboring subunit's C-terminal tail loop (Supplementary Fig. 10i). Moreover, the nicotinamide ring together with Tyr53 and His130 - two important residues in the catalytic tetrad[46] (Supplementary Fig. 3), form an anion hole where the substrate aldehyde binds to permit reduction.

The other set of purified state structure contains two complete octamers (16 subunits) in each asymmetric unit with two and four NADP(H)s all bound to the same tetrameric side, respectively (Supplementary Fig. 10c, d). Unlike the first purified state structure, the two tetramers in each octamer of this structure exhibit different but regular features. All subunits of cofactor-bound tetramers behave similarly to the first purified state structure in which the functional loops are clear and ordered (Fig. 2b and Supplementary Fig. 14). However, cofactor-free tetramers exhibit a higher degree disordered conformation in every subunit (Fig. 2c, Supplementary Fig. 13g–i and Supplementary Fig. 14), including loops for NADP(H)-binding (residues 21–28, 230–241), the loop for substrate binding (residues 87–95) and the loop where the catalytic residue Tyr53 is located in (residues 50–59). Moreover, the side chain of the catalytic residue His130, faces the opposite direction of the active center, and the side chain of Trp214 twists. We further confirmed these structural features is not resulted from crystal packing (Supplementary Fig. 12). This disordered conformation exhibits unsuitability in multiple functional structural modules such as cofactor binding, substrate binding, and catalysis.

Noteworthy, the disordered conformation actually represents an ensemble of various non-functional conformations in which the disordered loops in the crystal structure oscillate randomly, so we consider this conformation to be dormant, and the subunits of cofactor-bound tetramers belong to the active conformation with ordered and suitable structural features, and the multiple conformations of AKRtyl suggest conformational plasticity.

To summarize, the eight subunits in the first purified state structure all belong to the active conformations, while the second structure contains half active and half dormant conformations which coincidentally camped based on conformational states to form tetramers. Upon further analysis, in both structures, we found that the dormant tetramers have no NADP(H) bound but the active tetramers bound at least one NADP(H) (Supplementary Fig. 13c–e). This regular characteristic suggests that the binding of NADPH induces the transformation of the conformation of the entire tetramer to be active. This relationship between conformations is consistent with the MWC (concerted or symmetric) model in allostery that posits the existence of two pre-existing quaternary states - tensed (T) and relaxed (R), whose equilibrium shifts upon ligand binding[16,18,47,48]. In AKRtyl, the dormant conformation corresponds to T and the active conformation corresponds to R and the T to R transition is not a rigid body motion, but shifting the conformational ensemble from disorder to order.

However, to propose the MWC model, it is necessary to further determine the pre-existence of cofactor-independent T and R conformations in AKRtyl. To obtain the apo structure of AKRtyl, we remove the carried cofactors by denaturation and refolding. We successfully obtained two apo structures of AKRtyl without any cofactor (Supplementary Table 6). The first apo structure is similar to our previous second purified-state structure, with two octamers in an

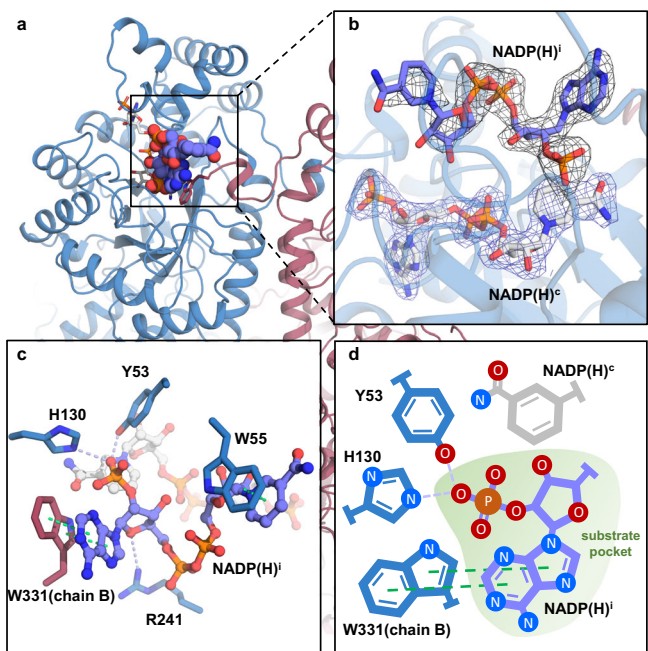

**Fig. 3 | AKRtyl binds two NADP(H)s and the NADP(H)$^i$ blocks the substrate binding site. a** Cartoon representation of the structure of AKRtyl and NADPH$^c$ (cofactor NADPH, white sticks), NADPH$^i$ (inhibitor NADPH, purple sphere) complexes. **b** Magnified section showing the two NADPHs. The structural model is superposed with the corresponding $2F_o$-$F_c$ electron density map contoured at $1.0\sigma$ (NADPH$^c$, blue; NADPH$^i$, black), the adenine group of NADPH$^i$ is clearly visible but the nicotinamide ring is indistinct. **c** Close-up view of the binding mode of NADPH$^i$ in the AKRtyl substrate pocket. The NADPH$^c$ is shown as white balls-and-sticks and NADPH$^i$ is shown as purple balls-and-sticks. Trp331 from the C-terminal tail loop forms π-π stacking interactions (green dotted lines) with the adenine moiety of NADPH$^i$, the Trp55 forms a π-π stacking interaction (green dotted line) with the nicotinamide ring and other residues form hydrogen bond interactions (light purple dotted lines) with the NADPH$^i$. **d** Schematic diagram showing the molecular mechanism of the binding of NADPH$^i$ robbing the substrate pocket and creating a not conducive conformation that inhibits the reaction.

asymmetric unit and one active ($R_4$) and one dormant ($T_4$) tetramer per octamer ($R_4T_4$) (Supplementary Fig. 13a). The second apo structure has only two active conformational (R) subunits in an asymmetric unit, by combining with other asymmetric units, we can obtain an octamer containing two active tetramers ($R_4R_4$) (Supplementary Fig. 13b). We didn't get the apo structures in the $T_4T_4$ state and speculate that structures in this state may not crystallize due to energetic issues. Further, we added 1.2-times NADPH during AKRtyl crystallization, and the structure shows that the octamer exhibits an all-liganded active state ($R_{4\text{-}N4}R_{4\text{-}N4}$) (Supplementary Fig. 13f). Combined with other liganded states ($R_{4\text{-}N2}T_4$, $R_{4\text{-}N4}T_4$, $R_{4\text{-}N1}R_{4\text{-}N1}$ as shown in Supplementary Fig. 13c–e), we find that the structure of the tetramers always exhibit symmetrical features.

Unlike the classical MWC model where different conformational equilibria occur in the actual oligomeric state, AKRtyl is actually octameric, but we observe symmetric T and R conformational equilibria within the tetramer. We therefore propose that the basic unit of MWC-type allosteric ensemble is a tetramer rather than an octamer for: (1) The tetramer exhibits ligand-independent T and R conformations, suggesting that the equilibrium between the T and R conformations occurs within the tetramer. (2) The tetramer has structural symmetry in different ligand states (unliganded, partially liganded, fully liganded). These features fit the MWC model[49]. Based on this, we propose that AKRtyl employs the MWC mode of allosteric regulation through NADPH binding. The dormant (T) conformation is favored at low NADP(H) concentration; the high NADP(H)-binding affinity for active

(R) conformation promotes a cooperative compulsory concerted conformational change driven by NADP(H) binding and involving all subunits within tetramers, displacing the equilibrium toward the R conformation. (Fig. 2e).

The purified state structures and apo structures revealed in detail how the enzyme binds cofactor NADPH and the intrinsic allostery which further promotes binding, but it's still confusing because the kinetic data indicated cofactor inhibition. Therefore, we cocrystallized AKRtyl with additional NADPH in a 1:10 (AKRtyl: NADPH) ratio. The AKRtyl·NADPH complex was diffracted to a resolution of 2.25 Å (Supplementary Table 6), with a minimal change in the overall octameric conformation compared to the purified-state structure (Supplementary Fig. 15a). The eight subunits all bind cofactors and behave R conformations unanimously (Fig. 3a). However, to our surprise, in addition to the classic NADP(H) binding mentioned above, we found that all subunits in the octamer additionally bind another NADP(H) molecule (Fig. 3b and Supplementary Fig. 16). Unlike the cofactor NADP(H) (abbreviated as NADP(H)$^c$ in structure), this NADPH presents a compacted conformation and is bound in the active pocket, occupying the position for substrate binding like an inhibitor (abbreviated as NADP(H)$^i$ in structure). Intriguingly, the substrate binding pocket seems to be tailored for NADP(H)$^i$ (Fig. 3c), where Trp331 from the C-terminal tail loop of the neighboring chain forms π-π stacking with the adenine moiety of NADP(H)$^i$. Noteworthy, Trp55 deflects compared to other structures, and it appears that NADP(H)$^i$ promotes its motion and has a relatively weak interaction with it. In addition, several other residues also interact with NADP(H)$^i$, including Tyr53 and His130 that form hydrogen bonding with the 2'-phosphate moiety of adenosine side, as well as hydrophobic interactions. All these interactions contribute to a stable occupancy for NADP(H)$^i$ in the substrate binding pocket. Binding of NADP(H)$^i$ robs the active site for substrate and prevents catalysis (Fig. 3d). This explains the high concentration inhibitory effect of NADPH on AKRtyl.

As demonstrated above, the nicotinamide ring of NADP(H)$^c$ and the adenine ring of NADP(H)$^i$ form π-π stacking with Trp214 and Trp331, respectively (Supplementary Fig. 10i and Fig. 3c), which implies that binding of NADPH is associated with intrinsic tryptophan fluorescence quenching. Thus, we monitored NADPH binding in real-time using stopped-flow spectroscopy. We found that fluorescence kinetic transients fitted better to a double-exponential decay rather than to a mono-exponential one. The differences in the apparent rate constants $k_{obs}$ between the fast and slow phases were ~5-fold (Fig. 4a, Supplementary Fig. 17 and Supplementary Table 7). Next, we measured the AKRtyl·NADPH interaction by determining the fluorescence quenching ratio in equilibrium in solution. As expected, the data fitted better to the two-step model rather than the one-step one (Fig. 4b, Supplementary Fig. 18a and Supplementary Table 9). On the contrary, the alanine mutant of residue Trp331, which plays a role in the binding of both NADPHs, exhibits significantly reduced affinity for both NADPH$^c$ and NADPH$^i$, and the data shifted towards the one-step model (Supplementary Fig. 18b, Supplementary Table 9). In addition, the fluorescence quenching trances of W331A fit the mono-exponential decay equation very well, while the double-exponential decay equation can only be fitted at high NADPH concentrations, but the quenching amplitude and quenching rate of the second process are weak (Supplementary Fig 19 and Supplementary Table 8). Accordingly, the NADPH inhibitory effect was almost relieved on the W331A mutant (Table 1 and Supplementary Fig. 20).

These results indicate that binding of NADPH to AKRtyl involves at least two processes that quench the intrinsic tryptophan fluorescence and confirm the binding of NADPH$^i$ to AKRtyl in solution, and the inhibitory binding corresponds to the second process of fluorescence quenching and is related to residue Trp331. Moreover, based on the MWC allosteric regulation in the cofactor NADPH binding process mentioned above, we introduce the Hill coefficient for the first

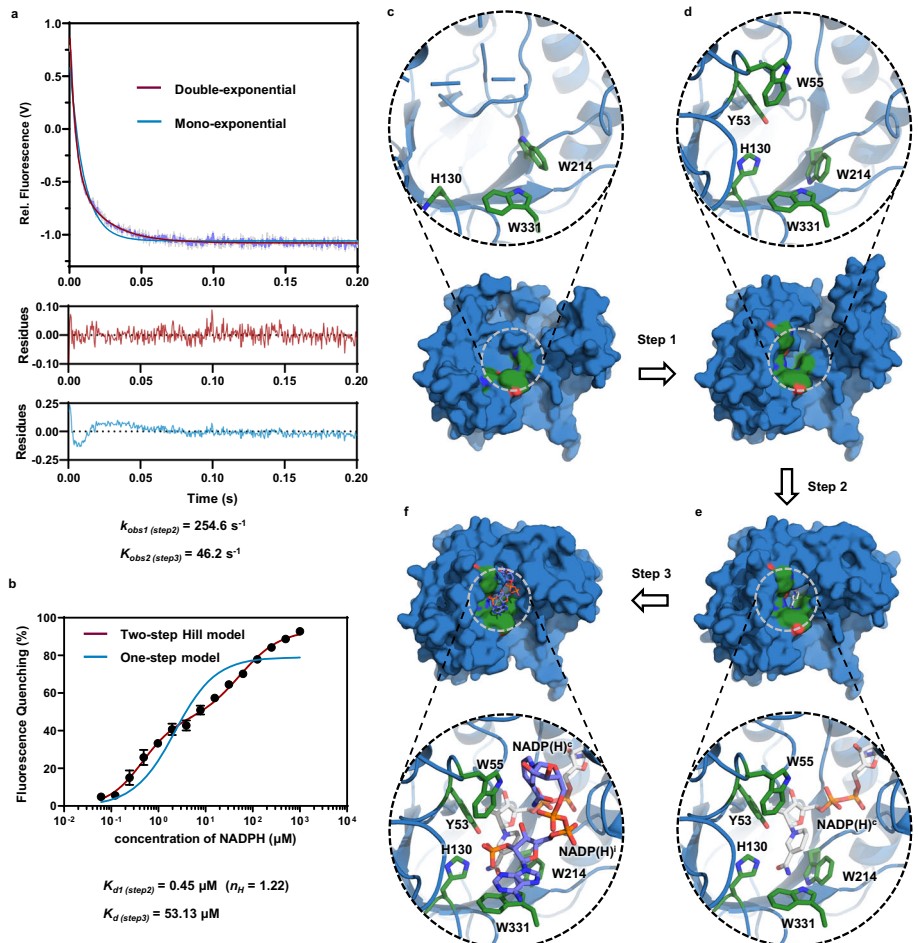

$k_{obs1 (step2)}$ = 254.6 s$^{-1}$

$K_{obs2 (step3)}$ = 46.2 s$^{-1}$

$K_{d1 (step2)}$ = 0.45 μM   ($n_H$ = 1.22)

$K_{d (step3)}$ = 53.13 μM

**Fig. 4 | The complete three-step model for NADPH binding to AKRtyl. a** Time course for binding NADPH (50 μM) to AKRtyl (0.2 μM) measured by monitoring the tryptophan fluorescence change quenched by NADPH. The purple line represents the average of three consecutive trials and the gray area represents the SD ($n = 3$ independent experiments). The data fit well to a double-exponential decay (red line; $R^2 = 0.9919$) rather than to a mono-exponential decay (blue line; $R^2 = 0.9671$). The residuals are shown below the fluorescence transient curve. Fitting to double-exponential decay yielded two $k_{obs}$ of 254.6 s$^{-1}$ and 46.2 s$^{-1}$. **b** Curve of AKRtyl's fluorescence quenching ratio with NADPH concentration at equilibrium. The data fit well to a modified two-step Hill model (red line; $R^2 = 0.9591$) rather than to a one-step model (blue line; $R^2 = 0.8783$) and two $K_d$ values of approximately 0.45 μM and 53.5 μM, the Hill value ($n$) of the first step is 1.22 (>1), indicating positive cooperativity. Error bars indicate means ± SD ($n = 8$ independent experiments). **c–f** The complete three-step for NADPH binding. **c** The dormant conformation. **d** The active conformation. **e** Binding of the NADPH$^c$. **f** Binding of the NADPH$^i$ in the substrate pocket.

step in the two-step model used in fluorescence quenching assays in the equilibrium solution state. The data fitted well with $K_{d1}$ of 0.45 μM and $K_{d2}$ of 53.5 μM, and the Hill value ($n$) in the first step is 1.22 (Fig. 4b, Supplementary Fig. 18b and Supplementary Table 9), further confirming the existence of positive allosteric regulation in the NADPH$^c$ binding process. And in the transient fluorescence quenching kinetics, we think that $k_{obs1}$ corresponds to the cooperative binding of NADPH. By fitting these data with conformational selection (the simplest form of the MWC) and induced fit (the simplest form of the KNF) models[50], we found that the former is more appropriate, as the fitted apparent dissociation constant $K_{d,app}$ (0.79 μM) is closer to the $K_{d1}$ (0.45 μM) (Supplementary Fig. 21).

Thus far, we have found a two-level regulation of NADPH by AKRtyl: (1) allosteric regulation of NADPH at micromolar level concentrations, inducing a shift in AKRtyl from a dormant to an active conformation which employs the MWC allosteric model, and (2) cofactor inhibition at sub-millimolar concentrations by "robbing" the substrate pocket. We then wondered how these regulations of NADPH affected the equilibrium of the order bi-bi reaction. Therefore, we determined the equilibrium constants of the reaction at different NADPH concentrations. We found that the equilibrium constant $K_{eq}$ exhibits an increase and then a decrease as a function of NADPH

concentration (Supplementary Table 5), e.g., $K_{eq} = 0.86$ when [NADPH] = 5 μM, and $K_{eq} = 81.37$ when [NADPH] = 50 μM, an almost 100-fold (2 orders of magnitude) increase. This means that NADPH pushes the equilibrium towards reduction within a certain concentration range. And when [NADPH] = 200 μM, $K_{eq} = 0.89$, which is nearly 100 times (2 orders of magnitude) lower compared to [NADPH] = 50 μM. This means that by further increasing the NADPH concentration, the equilibrium is pulled back from the reduction. And this is consistent with the two-level regulatory mechanism of NADPH, i.e. at micromolar level concentrations NADPH induces a conformational shift of AKRtyl from dormant to active conformations, favoring the reaction, whereas at sub-millimolar concentrations NADPH acts as an inhibitor, binding to the substrate pocket and inhibiting the reaction.

Combining different conformational states observed in the crystal structures, we propose a three-step model for NADPH binding which involves four conformational ensembles. The whole process begins with a dormant conformation prior to cofactor NADPH binding (Fig. 4c), which is holistically shifted to the active conformation (Fig. 4d) through ordering of functional loops and key residues. This process is under allosteric regulation of NADPH, consistent with the MWC model. The second step is that AKRtyl binds the cofactor NADPH

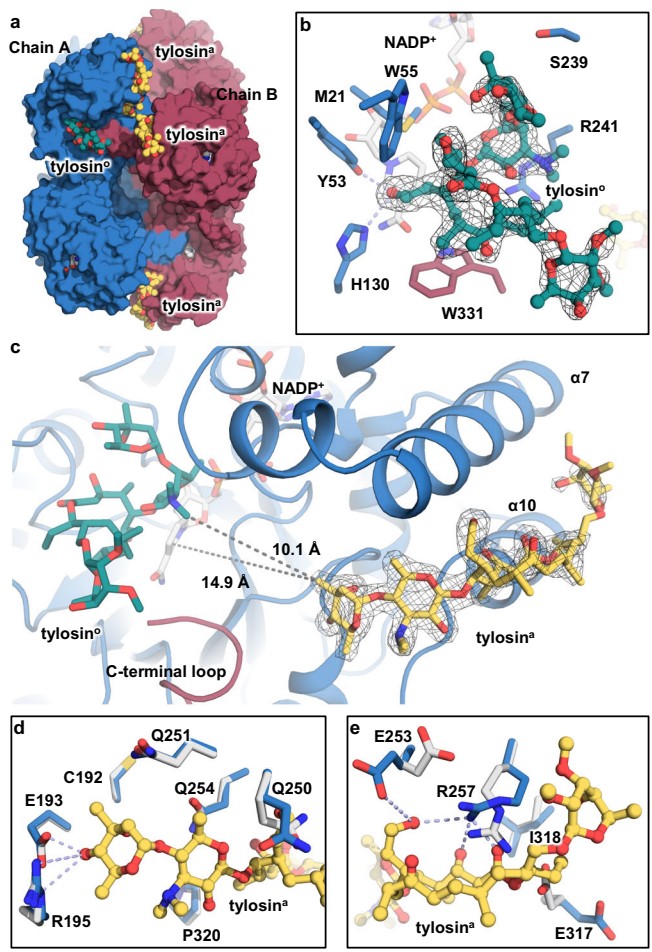

**Fig. 5 | AKRtyl binds two tylosins and tylosin[a] functions as allosteric modulator. a** Side views of the AKRtyl octamer binding to two tylosins (tylosin[o]: orthomeric tylosin, green sphere; tylosin[a]: allosteric tylosin, yellow sphere). **b** Magnified section showing the tylosin[o] and its binding site. The tylosin[o] is shown as green balls-and-sticks, the interacting residues are shown as blue or red (another chain) sticks, and the NADP[+] is shown as white balls-and-sticks in the back. The tylosin[o] structural model is superposed with the corresponding $2F_o$-$F_c$ electron density map contoured at $0.7\sigma$ (black). Tyr55 and His130 form hydrogen bond interactions (light purple dotted lines) with the aldehyde group of the substrate. **c** Close-up view of the binding mode of the tylosin[a] on the allosteric site (α-helices 7 and 10). The tylosin[a] is shown as yellow balls-and-sticks. The structural model is superposed with the corresponding $2F_o$-$F_c$ electron density map contoured at $1.0\sigma$ (black). The closest distance from the tylosin[a] to the tylosin[o] and NADP[+] are 10.1 Å and 14.9 Å, respectively. **d**, **e** The disaccharide side and macrolide side of the allosteric site and the superposition of the tylosin[a] bound (blue) and unbound (gray). The interacting residues are shown as sticks and the hydrogen bonding interactions are shown as light purple dotted lines. Glu193 and Arg195 forming hydrogen bonding interactions with the hydroxyl group at the C4 position of α-L-mycarose and Glu253 and Arg257 with the aldehyde group at the C20 position on tylosin and others are mainly non-polar hydrophobic interactions.

and form the enzyme-cofactor complex (Fig. 4e). Finally, with excess NADPH, the inhibitor NADPH binds to the substrate pocket and forms a non-reactive ternary complex (Fig. 4f). Our proposed three-step NADPH binding model for AKRtyl integrates the productive NADPH binding, as well as the two regulatory steps (MWC-type allostery and high concentration inhibition). This differs from the three-step model of other AKRs, such as AKR1C9[41] (Supplementary Fig. 22).

## Substrate tylosin functions as a positive allosteric modulator

After figuring out why excess cofactors inhibit AKRtyl, we turned to probe the molecular basis of its positive synergy on substrates. We

obtained AKRtyl ternary complex with NADP[+] and tylosin at 1.93 Å by cocrystallizing AKRtyl in the presence of cofactor and substrate at a molar ratio of 1:1.2:10 (AKRtyl: NADP[+]: tylosin). The structure of this complex is similar to the AKRtyl-NADPH binary complex (Supplementary Fig. 11b). In the ternary complex, we found an incredible phenomenon that one AKRtyl monomer binds two molecules of tylosin - one at the substrate binding site where the reaction happens and the other at a remote site which is 15 Å away from the active site (Fig. 5a, c).

The first tylosin (orthosteric tylosin, abbreviated as tylosin[o]) at the active site is refined with an occupancy of 79% and the position is different compared with that of the second NADPH in the AKRtyl-NADPH binary complex, thus excluding the possibility of NADP[+] binding (Fig. 5a, b and Supplementary Fig. 25). Tylosin[o] is close to NADP[+] with its C20 aldehyde carbon, the one to be reduced, 3.8 Å from the *para*-carbon of the cofactor's nicotinamide ring. The distance between the aldehyde oxygen and the phenolic hydroxyl of Tyr53 is 3.0 Å. This conformation satisfies the general "push-pull" mechanism of AKRs[51] in which the protonated Tyr53 (TyrOH$_2^+$) acts as the general acid by participating in the proton relay with His130 imidazole group to polarize tylosin's aldehyde group to accept a hydride ion from NADPH (Supplementary Fig. 26). Consistently, mutant Y53F completely inactivated AKRtyl and the catalytic efficiency of H130A decreased 50-fold (Supplementary Table 2 and Supplementary Fig. 27). Tylosin[o] at the active site makes contact with several other residues which are responsible for substrate binding and accessibility, including Trp55, Ser239 and Arg241. Mutating these residues exhibit varying effects on binding affinity and catalytic rate (Supplementary Table 2 and Supplementary Fig. 27).

Outside the active site, the second tylosin binds above the two extra helices α7 and α10 which lay outside the TIM barrel. It has slightly higher occupancy of 93% with well-defined electron density except for the glycosyl β-D-mycinose (Fig. 5c, Supplementary Fig. 24). In addition, out of the eight subunits in one octamer, six subunits bind tylosin while the other two is blocked by adjacent asymmetric units due to crystal packing. The distance between the C20 aldehyde on the tylosin and the *para*-carbon of the nicotinamide ring of NADP[+] is 25 Å thus its reduction is impossible. This remote binding site is reminiscent of another allosteric mechanism that ligand binding at allosteric sites spatially and topographically regulates macromolecule activity (so we call this tylosin as allosteric tylosin, abbreviated as tylosin[a]). The remote tylosin[a] binding site revealed that there were relatively few polar interactions (Fig. 5d, e), with hydrogen bonding between Glu193 and Arg195 and the C4 hydroxyl of α-L-mycarose, as well as Glu253 and Arg257 and the C20 aldehyde on tylosin[a]. The rest were essentially nonpolar hydrophobic interactions, consistent with the feature that allosteric sites exhibit more hydrophobic interactions[52–54].

To test the hypothesis that tylosin[a] at the remote site affects AKRtyl through allostery, we generated several mutants that either weaken or disrupt the binding (Table 1 and Supplementary Fig. 28). As expected, mutants E193A, E193W, R195W, and R257W almost lost the positive allosteric effect ($n = 0.95 \sim 1.08$), while R195A, R257A, and Q254W exhibits decreased cooperativity ($n = 1.12 \sim 1.38$). Changes at the allosteric site seem to affect tylosin binding at the active site, as all tested mutants showed reduced affinity (higher $K_d$ measured in single turnover experiments, Supplementary Fig. 29). Therefore, we concluded that the tylosin[a] binding site is a functional allosteric site, which will modulate the catalytic performance at the active site. Further, we determined the equilibrium constants of the reaction at different tylosin concentrations to investigate the effect of this allosteric regulation on the equilibrium of the reaction. We find that $K_{eq}$ increased almost 20-fold when tylosin increased from the micromolar (25 μM) to the submillimolar level (100–500 μM), whereas a further increase in tylosin concentration (1 mM) resulted in a smaller change in $K_{eq}$, which was in the same order of magnitude (Supplementary Table 5). This

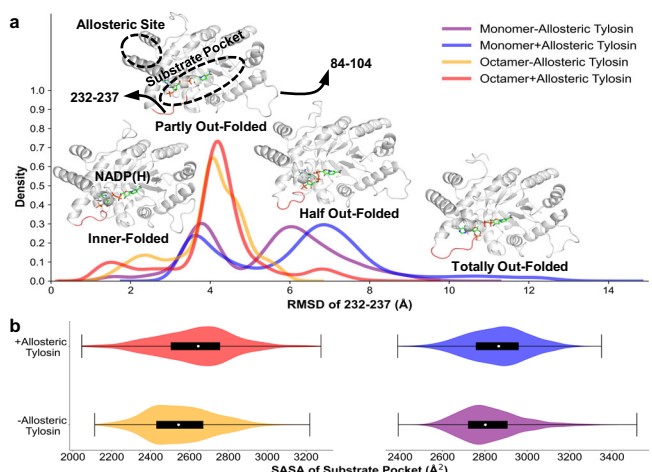

**Fig. 6 | Allosteric tylosin promotes the binding of orthosteric substrate.**
Molecular dynamics simulations of monomeric binding (purple) and non-binding (blue) allosteric tylosin and octameric binding (red) and non-binding (yellow) allosteric tylosin. **a** The conformation of AKRtyl-NADP(H) (white in cartoon) is mainly clustered into four categories based on loop 232–237 (red in cartoon) during simulations. At the monomer level, the conformations are mainly partly out-folded or half out-folded, after allosteric binding, the ensemble becomes more open with totally out-folded state emerging and no inner-folded conformation. At the octamer level, the conformations are mainly partly outer-folded, and the allosterically bound ensemble has additional accessible half out-folded conformations. **b** Calculation of the solvent-accessible surface area (SASA) of the substrate binding site during simulations. (Monomer-Allosteric Tylosin, $n = 1500$; Monomer +Allosteric Tylosin $n = 758$; Octamer-Allosteric Tylosin, $n = 1500$; Octamer +Allosteric Tylosin $n = 1500$). On violin plots, the upper and lower lines define the range of values while the box defines values between 0.25 and 0.75 quartiles; the white dot points the median. The substrate binding pocket was larger with a higher SASA both after allosteric tylosin binding at both monomer and octamer level.

implies that tylosin is able to push the equilibrium toward reduction. This is also consistent with our observation that tylosin acts as an allosteric modulator to facilitate catalysis.

Beyond that, we also obtained a binary AKRtyl structure in complex with tylosin at 2.25 Å resolution. The overall construction of this structure is similar to the the first apo structure and all 16 subunits are in the cofactor-free state. However, the tylosin molecule is bound to each chain at the allosteric site consistent with that observed in the ternary complex structure (Supplementary Fig. 30). Intriguingly, each octamer in the structure has half dormant and half active conformation which are grouped into two tetramers, consistent with the aforementioned MWC allosteric model. This implies that binding of allosteric tylosin is independent of cofactor binding and irrelevant to the conformational states of AKRtyl.

Taking together, the structural and kinetic results revealed that a second tylosin binds to an allosteric site and acts as a positive modulator. However, except for repositioning of a few residues which are hydrogen-bonded with tylosin[a], we were unable to observe significant conformational change. Therefore, we assumed that the allosteric mechanism may not involve large conformational change but only causes subtle behavior which may affect protein dynamic motion.

**Allosteric tylosin promotes the binding of orthosteric substrate**
To further explore the underlying mechanism of allosteric binding of tylosin that boosts the catalysis of AKRtyl, allosterically bound and non-bound AKRtyl-NADP(H) monomer were simulated for 1 μs and 3 rounds (Supplementary Fig. 31a). Principle component analysis (PCA) of trajectories suggested that the swing of two loops (residue 84–104 and residue 231–237) and C-terminus accommodate dominant motions (Supplementary Fig. 31b). The two loops, which are parts of

the substrate pocket as mentioned above, were previously validated crucial in the substrate binding and catalysis in AKR superfamily[46]. Conformation clustering of trajectories was then performed using the $C_\alpha$-RMSDs on the two loops (Supplementary Fig. 31c) and in consequence, four conformations were observed, split mainly by RMSD of loop 231–237 (Fig. 6a). With the increase of RMSD, loop 231–237 is converted from an inner-folded conformation to an out-folded conformation. In both ensembles, the loop is mainly partly out-folded or half out-folded. However, after allosteric binding, a totally out-folded state emerges (~3.5%) while no inner-folded conformation (<1%) is found for the loop 231–237. The increased degree of the out-folding of the substrate-binding loop would promote the ability of AKR in contact with substrates flowing in solvent. The promotion was further validated by calculation of solvent-accessible surface area (SASA)[52,55] of the substrate pocket, that allosteric binding has resulted in a larger pocket (Fig. 6b) so that substrate binding is kinetically easier. In addition, since AKRtyl is functional in octamer, we also simulated the whole octamer both with and without tylosin[a] binding at the allosteric site for 200 ns (Supplementary Fig. 31a) to see whether the conformation alternation also exists. The motion of loops in octamers is restricted, especially for 84–104 (Supplementary Fig. 31b), possibly due to the crowded environment after polymerization. In both ensembles, loop 231–237 is mainly partly outer-folded, while in allosteric-bound AKRtyl, the loop additionally has 7.6% of conformations in half out-folded state. The substrate binding pocket was also found larger with a higher SASA. Therefore, the same conclusion is drawn in both models that allosteric binding of tylosin to AKRtyl would promote the out-folding of the substrate binding loop and enlarges the surface area of the substrate binding pocket, leading to an easier binding of substrates in a kinetic aspect. The above observations were found equilibrated among the last half of the trajectories (Supplementary Figs 32–35). We have also examined the residue-level mechanism for this allosteric regulation. The remote control was predicted along a residue pathway of R257 → Q256 → I226 → E229 → Q230 → G232 → G233 → N234, via a short α-helix (225–230) under allosteric site towards loop 84–104 (Fig. 6a).

**The conservative three-level regulatory mechanism in AKR superfamily**
Of the 3898 sequences in AKR family 12, 1357 share more than 60% identity with AKRtyl, accounting for more than 1/3. SSN analysis also shows that the cluster where AKRtyl is located has the most nodes (Supplementary Fig. 36a), suggesting that the D subfamily may be the largest in family 12. Of the 1357 sequences in the D subfamily, 710 sequences are from *Streptomyces*, accounting for more than half, and over 75% of these *Streptomyces* AKRs and AKRtyl have a sequence identity exceeding 80% (Supplementary Fig. 36b), indicating that AKRtyl represents a broad cluster of highly conservative AKRs from *Streptomyces* which are major sources of natural products and produce 70–80% bioactive compounds with pharmacological or agrochemical applications[56,57].

Multiple sequence alignment of AKRtyl with several AKRs from model *Streptomyces* revealed that the C-terminal tail loop (including Trp331), the cofactor binding site, the substrate binding site (also NADPH[i] binding site), and the region interacting with α-L-mycarose of tylosin[a] (Glu193 and Arg195) are highly conserved in AKR12D subfamily (Supplementary Fig. 36c). This implies that in these *Streptomyces* AKRs, the three-level regulation mechanism observed in AKRtyl may be conservative. Therefore, we measured the kinetic properties of two AKRs from *S. coelicolor* A3(2) and *S. albus* J1074. Steady-state kinetics revealed that they not only can reduce tylosin but also exhibit positive cooperativity, with Hill coefficients ($n$) of 1.41 and 1.50, respectively (Supplementary Table 4 and Supplementary Fig. 37a, b). In addition, inhibition by the cofactor NADPH at high concentrations was also observed for these two AKRs (Supplementary Table 4 and

Supplementary Fig. 37c, d). These results suggest that the multiple level properties on cofactor and substrate binding of AKRtyl may be a common phenomenon in *Streptomyces* AKRs.

Further zooming in the entire AKR superfamily, we recognize that AKRs are distributed throughout kingdoms of life and share a similar broad substrate profile with conserved monomeric structural features. To investigate whether they have similar substrate synergy compared to AKRtyl, we turned to AKRs from other organisms, including AKR1A1 from the eukaryotic vertebrate *H. sapiens*[58], AKR4C9 from the eukaryotic plant *A. thaliana*[59], AKR3A2 from the eukaryotic microorganism *S. cerevisiae*[60] and AKR5C2 from the prokaryotic microorganism *E. coli*[61]. Although belonging to different AKR families, all four proteins were reported to display a broad substrate spectrum for aldehydes. Steady-state kinetics showed that these AKRs exhibited positive cooperativity for representative substrates ($n = 1.23 \sim 1.92$) (Supplementary Table 4 and Supplementary Fig. 38). Therefore, we speculate that positive substrate synergy may be a general property for AKRs from three kingdoms of life.

## Discussion

In this study, we have demonstrated an example of three-level structural regulation of tylosin reductase AKRtyl, which involves two distinct allosteric and one inhibitory mechanisms in single enzymatic system. This system involves the binding of multiple copies of NADP(H) and tylosin, leading to either noticeable or minute conformational changes with variable consequences for the reaction. Specifically, at lower cofactor concentrations, NADPH binding shifts AKRtyl conformational equilibrium from the dormant towards the active side which further promotes NADPH binding. What's special is that the MWC model-based allostery occurs in the tetrameric unit, even though the protein behaves as an octamer. Unlike the rigid body motions in quaternary structures induced by ligands of many other classical MWC models, such as haemoglobin[48,62,63], AKRtyl adopts a mode of cofactor-coupled shifting the conformational ensemble from disorder to order. This disorder-to-order activation pattern is similar to the allosteric regulation driven by intrinsically disordered regions[15,64], broadening the MWC model. Moreover, another NADPH occupies the substrate binding site when excess cofactor is present, causing an inhibitory behavior. In contrast to the "on/off" logic of conventional allosteric proteins, two-level regulation actions enable AKRtyl graded (rheostat-type) responses to NADPH. This pattern is similar to the two-level allostery of chaperonin GroEL which structurally behaves as a tetradecamer composed of two heptameric rings[65], the first level being an MWC-type allostery within the heptameric ring leading to ATP positive cooperativity, whereas the second level being a KNF-type allostery between the heptameric rings leading to negative cooperativity. Similar to AKRtyl, MWC allostery in GroEL is present in a separate heptameric ring, despite that the overall structure is tetradecameric[66,67]. We hypothesize that the existence of inter-ring synergies in the overall structure is related to the energy state and the inter-ring communications. From substrate point of view, binding of tylosin at the remote allosteric site promotes the binding of the orthosteric tylosin and enlarges the substrate pocket with microscopic conformational changes. And this type of cooperativity with minor conformational change is similar to other examples of allosteric stabilisation[68]. This enriches the profile of the regulatory mechanisms in AKR family, and the NADPH-independent mode of tylosin binding broadens our understanding of AKR substrate loading.

In addition to tylosin, AKRtyl shows positive cooperativity with many other substrates. However, we have no idea about the mechanism of these substrates as they were not visible in the co-crystallization structure. Nevertheless, a general perspective can be proposed to explain the properties of AKRtyl with other substrates based on the allosteric mechanism of tylosin on AKRtyl. The allosteric tylosin binding site is located on the two extra α-helices (α7 and α10), which lie on the periphery of the octameric scaffold and may prioritize exposure to external perturbations. In addition, the two α-helices are in a junction position, with the α7 helix connected upstream to the long NADPH-binding loop and the α10 helix connected downstream to the C-terminal loop. These loops together form the active site and play an important role in cofactor and substrate binding. Therefore, the two α-helices may sense the substrate concentration and transmit a signal to the active site. On the other hand, aldehydes can easily interact with proteins to form acetal adducts with thiol and amino groups on proteins[46,69]. If other aldehyde substrates can perturb the sensor (α7/α10) by interacting with residues such as lysine, arginine and cysteine on the two helices, it may drive loop movement like the allosteric tylosin binding, thereby affecting the active site. This may explain why AKRtyl also shows positive synergy with many other substrates. It is worth noting that this structural feature is common in the AKR superfamily, albeit low sequence conservation (Supplementary Fig. 3), and the two α-helices (α7/α10) position similarly in all AKRs as in AKRtyl. Our results show that other AKRs from different families and biological kingdoms also show positive cooperativity with substrates. Therefore, we speculate that the two extra α-helices located outside the TIM barrel may act as general sensors for the substrates and modulate catalysis.

Many AKRs are thought to play an important role in detoxification because aldehydes have high intrinsic chemical activity and readily react with nucleophiles, which damages proteins, DNA and lipids[46,69]. Aldehydes are ubiquitous in cells[70], some of them are harmful while others are important for the normal physiological metabolism, e.g. glyceraldehyde 3-phosphate and erythrose 4-phosphate. In addition, the cofactor (NADPH) required for the reduction of aldehydes are essential for a large number of biochemical reactions, and maintaining the cellular redox balance is a basic requirement for living cells[71,72]. This led to the hypothesis that the complex regulatory mechanisms at the structural level confer AKRtyl the ability for its response to varied intracellular substrate and cofactor levels. In terms of cofactors, when NADPH is poor, AKRtyl is in a dormant conformation with low activity, allowing scarce NADPH to be used for sustaining physiological metabolism. When NADPH level increases, an active conformation suitable for catalysis emerges, while continued increases of NADPH level - implying high reduction level and low oxidative stress - the active pocket of AKRtyl will be sequestered by NADPH to prevent rapid reduction of useful aldehyde metabolites. In terms of substrate, low concentration aldehydes may not pose a threat and may be essential in cellular metabolism, so the enzyme catalysis is at low level. However, as aldehyde concentration increases, AKRtyl shows an increase in substrate sensitivity and activity, i.e. it can better handle the oxidative stress caused by aldehydes.

In summary, using biochemical, biophysical, structural and computational approaches, we have elucidated the intrinsic multiple allosteric regulation mechanism of AKRtyl, which shows distinct properties toward cofactor and substrate at the molecular level, including an unusual MWC model-based allosteric regulation at low concentrations but an inhibitory effect at high concentrations in response to NADPH, and a positive synergistic allosteric regulation caused by the cofactor-independent tylosin binding. This serves as a distinct example to understand the complex multiple allosteric systems in nature and provides guidance for the design and modification of allosteric proteins and drugs.

## Methods

### General materials, bacterial strains and plasmids

The strains, plasmids and PCR primers used in this study are listed in Supplementary Table 11 and Supplementary Table 12. Primer synthesis and DNA sequencing were performed by Tsingke Biotechnology Co. (Beijing, China). Fast digest restriction endonucleases and T4 DNA ligase were purchased from Thermo Fisher (Shanghai, China). The

Super-Fidelity DNA Polymerase, Gel DNA Extraction Kit, Plasmid Isolation Mini Kit, Bacterial DNA Isolation Mini Kit and ClonExpress MultiS / One Step Cloning Kit were purchased from Vazyme Biotech Co. (Nanjing, China). All these kits were used according to the manufacturer's procedures. Tylosin, NADPH and NADP⁺ were purchased from Macklin Co. (Shanghai, China). Other chemicals and media components were obtained from standard commercial sources and used directly.

### Cultivation, fermentation and measurements

*E. coli* strains containing plasmids were cultured in LB medium at 37 °C with appropriate antibiotic selection. *S. fradiae* TL-01 and mutant strains were cultured in liquid TSB medium for mycelium, solid IWL4 medium for sporulation and *E. coli-Streptomyces* conjugation. For fermentation of tylosin derivatives[73], *S. fradiae* TL-01 and mutant strains were germinated in 30 mL of seed medium (soybean meal 0.5%, yeast extract 0.3%, corn steep liquor 1.0%, calcium carbonate 0.2% and soybean oil 0.5%, pH = 7.0–7.2) at 30 °C, 220 r for 2 days, and then 1.5 mL of seed culture was transferred to 30 mL fermentation medium (corn powder 1.0%, corn protein 0.6%, fish meal 0.7%, calcium carbonate 0.3%, NaCl 0.05%, soybean oil 4%, betaine HCl 0.5%, CoCl₂ 0.0001%, pH 7.0–7.2) and incubated at 30 °C, 220 r for 7 days. The fermentation samples were centrifuged at 12,000 rpm for 10 min and the supernatants after being diluted 20-fold were used for HPLC analysis.

HPLC analysis was carried out on an Agilent (1260 series) equipped with a reversed-phase column (ACE Excel, C18, 5 μm, 4.6 × 150 mm) for monitoring enzymatic reactions and fermentation products with UV detection at 280 nm. The mobile phase consisting of 0.1% formic acid in water (A) and acetonitrile (B) with flow rate 0.6 ml/min, and the following gradient was used: 0–4 min, 25% B; 4–26 min, 25–40% B; 26–27 min, 40%-100% B; 27–32 min, 100% B; 32–33 min, 100–25% B; 33–40 min, 25% B. Tylosin was eluted at approximately 20 min, and relomycin eluted at 18 min. LC-MS was performed on an Agilent 1260 Infinity II/6545 QTOF (LC/MS) in positive ionization mode.

### Construction of mutant strains

To construct plasmids for in-frame deletion, two 2-kb fragments of the gene upstream and downstream of AKRtyl were amplified from the genome with the primers listed in Supplementary Table 12, respectively. Both fragments were cloned into the linearized plasmid pYH7 using Gibson assembly strategy and the resulting construct pYH7-AKRtyl was introduced into *S. fradiae* TL-01 by *E. coli-Streptomyces* conjugation. Single-crossover mutants of AKRtyl were selected by apramycin resistance and apramycin-sensitive double-crossover mutants were selected after several rounds of passaging the exconjugants.

### Gene cloning and site-directed mutagenesis

All protein coding genes (Uniprot ID: AKRtyl-A0A3R7J519; AKR1A1-P14550; AKR3A2-Q12458; AKR4C9-Q0PGJ6; AKR5C2-Q46857; AKR_coelicolor-Q9F2Z5; AKR_albus-A0A6B3HBY1) were either amplified by PCR or commercially synthesized and cloned into pET28a expression vector between *Nde*I and *Xho*I sites. Mutagenesis was carried out in inverse-PCR using Super-Fidelity DNA polymerase according to the manufacturer's protocol. Primer list is available in Supplementary Table 12. All plasmids were verified by sequencing.

### Gene expression and protein purification

Plasmids harboring target genes were transformed into *E. coli* BL21 (DE3) cells for protein production. Cells were grown in LB medium supplemented with 50 μg/ml kanamycin at 37 °C with shaking at 220 rpm until an OD600 of 0.6–0.8 was reached. Protein expression was induced by the addition of 0.15 mM isopropyl-β-D-1-

thiogalactopyranoside (IPTG) and the cells were further grown for 20 h at 18 °C with shaking. After harvesting by centrifugation at 4000 g for 10 min, cell pellets were collected and resuspended in buffer A (50 mM Tris, pH 7.5, 300 mM NaCl), lysed using high-pressure homogenizers (Union Biotech), and centrifuged at 17000 g (60 min, 4 °C) to remove the cell debris. The target protein in the supernatant was purified by nickel-affinity chromatography using gravity-flow columns packed with Ni-NTA resin (SMART life sciences) and further purified by size-exclusion chromatography using a Superdex 200 increase 10/300GL column (GE Healthcare) in buffer C (10 mM Tris, pH 7.5, 150 mM NaCl) on the AKTA pure system. Fractions containing the target protein were collected and concentrated using an Amicon® concentrator (Millipore). For cofactor-free AKRtyl proteins, the Ni-NTA resin with bound proteins was first washed with 5 column volumes of protein denaturation buffer (10 mM HEPES, 300 mM NaCl, 2 M KBr, 2 M CO(NH₂)₂) to remove the cofactors. The Ni-NTA resin was then washed with 20 column volumes of buffer A to refold the protein. Other steps were as described above. Protein concentration was determined based on the UV absorbance at 280 nm using the molar absorptivity constant calculated by Expasy (ε (AKRtyl) = 0.62 M⁻¹·cm⁻¹·Da⁻¹, ε (AKR1A1) = 0.65 M⁻¹·cm⁻¹·Da⁻¹, ε (AKR3A2) = 0.75 M⁻¹·cm⁻¹·Da⁻¹, ε (AKR4C9) = 0.61 M⁻¹·cm⁻¹·Da⁻¹, ε (AKR5C2) = 0.63 M⁻¹·cm⁻¹·Da⁻¹, ε (AKR14A1) = 0.69 M⁻¹·cm⁻¹·Da⁻¹, ε (AKR_coelicolor) = 0.72 M⁻¹·cm⁻¹·Da⁻¹, and ε (AKR_albus) = 0.74 M⁻¹·cm⁻¹·Da⁻¹)[74]. Finally, the purified proteins were flash-frozen in liquid nitrogen and stored at −80 °C until use.

### SEC-MALS

A Superdex 200 increase 10/300 GL column (GE Healthcare) was coupled to an AKTA FPLC system, a multi-angle static light scattering detector (miniDawn, Wyatt), and a differential refractive index detector (Optilab, Wyatt). The column was equilibrated with buffer C used in protein purification. 200–300 μl AKRtyl protein at 50 μM were filtered and loaded into the column. The elution profiles were analyzed using ASTRA 6 (Wyatt).

### Enzyme activity assay and kinetic studies

AKRtyl activity was measured by following the decrease of NADPH absorbance at 340 nm for 10 min using a BioTek Synergy H1 microplate reader. The assays were performed at 30 °C in 100 μl reaction mixtures consisting 10 mM Tris-HCl (pH 7.5), 150 mM NaCl, 1.5 mM NADPH, 1.0 mM substrates and 0.2 mg ml⁻¹ (~5 μM) AKRtyl. The control experiments were performed with boiled enzyme. The specific activity of AKRtyl was calculated from the maximum rate of the corresponding kinetic curves which were given by the instrument directly. Each assay was performed in triplicate.

All kinetic assays were performed by following the decrease of NADPH absorbance at 340 nm for 2 min as described above. Kinetics with respect to NADPH were studied with saturated tylosin (5 mM) and at variable concentrations of NADPH (from 0 to 200 μM). Kinetic parameters were obtained by fitting to the substrate-inhibition Eq. (1). Kinetics with respect to substrates were studied with saturated NADPH (200 μM) and at optimized variable substrate concentrations (tylosin: from 0 to 2.5 mM). Kinetic parameters were obtained by fitting to the allosteric-sigmoidal Hill Eq. (2).

$$v = \frac{V_{max}}{1 + \frac{K_m}{[S]} + \frac{[S]}{K_i}} \tag{1}$$

$$v = \frac{V_{max} \times [S]^n}{K_{0.5}^n + [S]^n} \tag{2}$$

where $v$ is the initial rate at NADPH or tylosin concentration [S], $V_{max}$ is the fitted maximum rate, $K_m$ is the Michaelis-Menten constant, $K_i$ is the inhibition constant, $K_{0.5}$ is the apparent Michaelis constant in Hill

equation and $n$ is the Hill coefficient of substrate binding as a measure of cooperativity ($n > 1$, positive cooperativity; $n = 1$, no cooperativity; $n < 1$, negative cooperativity).

## Protein crystallization and structure determination

Crystallization experiments were carried out by the vapor diffusion method using the sitting-drop technique at 20 °C. Drops were equilibrated against a reservoir filled with 0.5 ml well solution. Purified AKRtyl protein aliquots were diluted to 10 mg ml$^{-1}$ in 10 mM Tris-HCl, pH 7.5, 150 mM NaCl.

The purified-state crystals were obtained by mixing 1 μl of enzyme with 1 μl reservoir solution consisting of 0.1 M MES, pH 6.0 and 22% PEG 400. The apo crystals were obtained by mixing 1.5 μl of enzyme with 0.75 μl reservoir solution consisting of 0.1 M imidazole, pH 7.0 and 50% MPD. For the AKRtyl·NADP(H) complex, enzyme was mixed with NADPH at a molar ratio of 1:1.2 and 1:10 (AKRtyl: NADPH) and crystals were obtained by mixing 1.5 μl of the complex with 0.75 μl reservoir solution consisting of 0.1 M Sodium citrate, pH 5.5 and 18% PEG3350. For AKRtyl·NADP$^+$·tylosin ternary complex, the enzyme was mixed with NADP$^+$ and tylosin at a molar ratio of 1:1.2:10 (AKRtyl: NADP$^+$: tylosin) and crystals were obtained by mixing 1.5 μl of the complex with 0.75 μl reservoir solution consisting of 0.1 M Sodium citrate, pH 5.5 and 18% PEG3350. For AKRtyl·tylosin complex, enzyme was mixed with tylosin at a molar ratio of 1:10 (AKRtyl: tylosin) and crystals were obtained by mixing 1 μl of the complex with 1 μl reservoir solution consisting of 0.1 M MES, pH 6.0 and 22% PEG 400.

All crystals were fished out, cryogenically protected in the corresponding reservoir solutions supplemented with 10% glycerol, and flash-frozen in liquid nitrogen for data collection. X-ray diffraction data were collected under cryogenic conditions at 100 K at beamlines BL18U1 (wavelength 0.97915 Å) and BL19U1 (wavelength 0.97852 Å) at the Shanghai Synchrotron Radiation Facility (Shanghai, China) and processed using the HKL2000/HKL3000 software package[75] (HKL Research, Charlottesville, VA). MtmW[24] (PDB ID: 6OVQ) was used as the search model in the molecular replacement in the PHENIX PHASER module[76]. Refinement of the structures was performed using PHENIX REFINE[77] with 5% random reflections for the validation of the $R_{\text{free}}$ value[78] throughout the refinement process. COOT[79] was used between refinement rounds for manual building and corrections of structural models. Restraints for ligands were generated using the PHENIX ELBOW module[77]. Validation of the refined models was performed using the PDB validation server. Figures were generated with Pymol (Schrödinger).

## Sequence alignment and phylogenetic analysis

The AKR sequences were obtained from the AKR superfamily database[22] (https://akrsuperfamily.org/) and GenBank by BLAST search. The phylogenetic tree was constructed by MEGA11[80] using the neighbor joining method and optimized online by Chiplot (https://www.chiplot.online/)[81]. Multiple-sequence alignment was constructed using Clustal Omega (https://www.ebi.ac.uk/Tools/msa/clustalo)[82] and optimized online using ESPript 3.0 (https://espript.ibcp.fr/ESPript/cgi-bin/ESPript.cgi)[83].

## Generation of SSNs

The dataset for complete AKR SSN was collected from the Pfam database[84] (http://pfam-legacy.xfam.org/; accession number: PF00248), the Interpro database[85] (https://www.ebi.ac.uk/interpro; accession number: IPR023210), and the AKR superfamily database and contained approximately 330,000 sequences. The dataset for the family 12 AKR SSN was collected from the Interpro database (accession number: cd19087) and contained 3898 sequences after removal of duplicates. The Enzyme Function Initiative Enzyme Similarity Tool (EFI-EST)[40] (https://efi.igb.illinois.edu) was used to perform an all-by-all BLAST analysis of the representative sequence dataset described

above to generate an initial SSN with an alignment score threshold of 88 for complete AKR SSN and 142 for the family 12 AKR SSN. All networks were visualized with the Organic layout in Cytoscape[86].

## NADPH binding measured by stopped-flow fluorescence kinetics

AKRtyl (0.2 μM) was equally mixed with varying concentrations of NADPH at the buffer (10 mM Tris-HCl, pH 7.5, 150 mM NaCl) on a SX20 stopped-flow spectrometer (Applied Photophysics Ltd., UK). All spectra were collected with 290 nm excitation and a 310 nm long-pass filter at 30 °C. Averaged data from over 3 replicates were fitted by GraphPad Prism 8. The observed rate constant $k_{obs}$ was calculated using a single exponential Eq. (3) and a double exponential Eq. (4):

$$F_t = \Delta F e^{-k_{obs}t} + F_{eq} \tag{3}$$

$$F_t = (\Delta F)_1 e^{-k_{obs1}t} + (\Delta F)_2 e^{-k_{obs2}t} + F_{eq} \tag{4}$$

where $F_t$ is the fluorescence at time $t$, $\Delta F$ is the fluorescence amplitude decrease, $k_{obs}$ is the apparent first-order rate constant, and $F_{eq}$ is the fluorescence at equilibrium.

## NADPH binding measured by equilibrium fluorescence quenching

NADPH binding in the equilibrium solution state was determined experimentally using a BioTek Synergy H1 microplate reader. AKRtyl (0.4 μM) was mixed 1:1 with varying concentrations of NADPH in binding buffer (10 mM Tris-HCl, pH 7.5, 150 mM NaCl) to 40 μl in a 384-well microplate. Binding reactions were prepared in octuplicate. After incubation at 30 °C for 3 min, the fluorescence intensity was recorded under the condition of $\lambda_{\text{excitation}} = 290$ nm, $\lambda_{\text{emission}} = 340$ nm. Dissociation constants were calculated using a one-step (5), two-step (6), and two-step Hill (7) model:

$$F_L = F_0 + \frac{F_{\text{max}}[L]}{K_d + [L]} \tag{5}$$

$$F_L = F_0 + \frac{F_{\text{max}1}[L]}{K_{d1} + [L]} + \frac{F_{\text{max}2}[L]}{K_{d2} + [L]} \tag{6}$$

$$F_L = F_0 + \frac{F_{\text{max}1}[L]^n}{K_{d1}^n + [L]^n} + \frac{F_{\text{max}2}[L]}{K_{d2} + [L]} \tag{7}$$

where, $[L]$ is NADPH concentration, $F_L$ is the fluorescence quenching ratio at $[L]$, $F_{max}$ is the maximum fluorescence quenching ratio at saturation with binding constant $K_d$, $n$ is the Hill coefficient as a measure of cooperativity, and $F_O$ is the fluorescence quenching ratio without NADPH (so $F_O = 0$).

## Measurement of equilibrium constant

The equilibrium constant was determined based on the concentration of each component when the reaction reached equilibrium. Experiments were performed at different NADPH and tylosin concentrations. We varied the concentration of NADPH or tylosin and fixed the concentration of the other to study the effect of a single factor (The ligands concentrations used in experiments are shown in Supplementary Table 5), and the enzyme concentration was 1 μM. The assays were performed at 30 °C in 100 μl reaction mixtures consisting 10 mM Tris-HCl (pH 7.5), 150 mM NaCl. The 340 nm absorption was measured by a BioTek Synergy H1 microplate reader to detect whether the reaction had reached equilibrium and calculate the concentrations of reactants and products at each equilibrium end-point permitting the calculation of $K_{eq}$. Each assay was performed in triplicate.

## Molecular dynamics simulation

Four models were constructed, which are AKRtyl·NADP(H) monomer or octamer binding or without binding tylosin and at allosteric site. The solved crystal structure of AKRtyl·NADP(H) octamer with allosteric tylosin and without allosteric tylosin were used for construction of simulation systems. Monomer models were extracted from octamer. (More information in Supplementary Table 10) For all systems, the protein is described by ff14SB forcefield[87] while small molecules (NADP(H) and tylosin) are described by general Amber forcefield (GAFF)[88]. Partial charges of small molecules were assigned using RESP approach[89,90]. Missing atoms, contour ions and TIP3P solvation box were added, and input files were prepared using *tleap* module in AMBER20[91] package. All prepared systems were first performed with energy minimization and then heated under constant volume from 0 K to 300 K for 100 ps. The initial speeds of atoms are generated using the wall-clock time as the random seed for 3 rounds. After that, the complexes were equilibrated by two rounds of 100 ps simulation, each by NVT and NPT ensemble. Productions by NPT ensemble were performed for 1 μs and 3 rounds for monomer models while 1 round and 200 ns for octamer models. All-atom simulations with explicit solvent model and fixed charges under microsecond timescale are believed to be enough for studying conformation changes of loops in proteins[92]. All Langevin dynamics[93] were performed with a collision frequency of 2.0 ps under 300 K and standard pressure. Particle Mesh Ewald method[94] were used to calculate long-range electrostatic interaction, where the cutoff value was set at 8.0 Å. SHAKE algorithm[95] was used to constrain chemical bonds linking hydrogens. The integration step was chosen to be 0.5 fs in heating but switching to 2 fs in equilibration and production. All simulations were performed using CUDA accelerated *pmemd*[96] in AMBER20 package.

Analysis of trajectories, including principle component analysis (PCA), calculation of RMSD and calculation of motion covariance among $C_\alpha$ atoms (for calculation of regulation pathway) were performed with VMD 1.9.3 (LINUXAMD64)[97] and Prody 2.4.0[98] package. Calculation of SASA was done by Pymol 2.4.1. Regulation pathways were predicted under following workflow[92]. The protein was simplified to a graph, where $C_\alpha$ stand as nodes. An edge was added to two nodes only when the distance between their $C_\alpha$s was below 8 Å for at least 75% of the conformations in ensembles. The length of the edge linking node *i* and *j* was defined as below:

$$d_{ij} = -\log(c_{ij}) \tag{8}$$

where $c_{ij}$ is the motion covariance between node *i* and *j*. Regulation pathway was defined as the shortest pathway on graph from R257 towards N234, where R257 has a strong polar interaction with allosteric tylosin and N234 is the middle of loop 231–237. The shortest pathway was solved by Dijkstra algorithm[99].

For models not binding allosteric tylosin, all frames recorded in production are used for analysis (50000 frames × 3 rounds for the monomer model and 10000 frames × 1 round for the octamer model). For models binding tylosin, however, the tylosin could disassociate during simulation. Therefore, we only use frames with tylosin binding for analysis (50000 frames + 23000 frames (0–460 ns) + 2800 frames (0–56 ns) for 3 rounds of the monomer model and 10000 frames × 1 round for the octamer model).

## Reporting summary

Further information on research design is available in the Nature Portfolio Reporting Summary linked to this article.

## Data availability

Data supporting the findings of this work are available within the paper and its Supplementary Information and Source Data files. X-ray crystallographic coordinates have been deposited in the Protein Data Bank (PDB) with the accession codes: 8XR2, 8XR3, 8JWL, 8JWK, 8XR4, 8JWN, 8JWM, 8JWO. Previously published structure cited in this study can be accessed using the PDB accession number 6OVQ. Source data are provided as a Source Data file. Other data are available from the corresponding authors upon request. Source data are provided with this paper.

## Code availability

The code for trajectory analysis and notebook as well as source data related to the MD simulations are available at https://github.com/JinyinZha/AKRtyl. Supplementary Data 1 encompasses the optimized structure and Gaussian input files of NADPH and tylosin. Supplementary Data 2 contains the initial structures used in the MD simulations and representative conformations of the trajectories.

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

## Acknowledgements

This work was supported by grants from the National Key Research and Development Program of China (2018YFA0901900 and 2023YFA0914200), National Natural Science Foundation of China (21632007 for S.L., 32271305 for S.D., 81925034 for J.Z.), "Major Project" of Haihe Laboratory of Synthetic Biology (22HHSWSS00001), and Program of Shanghai Subject Chief Scientist (21XD1401300). We thank the staffs from BL18U1/BL19U1 beamline of National Facility for Protein Science in Shanghai (NFPS) at Shanghai Synchrotron Radiation Facility, for assistance during data collection and the Instrumental Analysis Centre of Shanghai Jiao Tong University for making mass spectroscopy. We thank Professor Xiao He at East China Normal University for providing computational time.

## Author contributions

S.L., S.D., J.Zhang and Z.X. designed the research and the experimental strategy. Z.X. expressed, purified, crystallized and enzymatically characterized proteins with support from T.H., Q.L. X.W. J.Zhong and R.L.; Z.X. collected crystallographic datasets with support from T.H., X.Y., S.H. and J.Zheng; Z.X. refined, analyzed and interpreted the crystallographic data under the supervision of S.D.; J.Zha carried out the molecular dynamics calculations under the supervision of J.Zhang; Z.X., J.Zha, S.L., S.D., J.Zhang and Z. D. wrote the manuscript.

## Competing interests

The authors declare no competing interests.

## Additional information

**Supplementary information** The online version contains
Supplementary Material available at

Jian Zhang, Shuangjun Lin or Shaobo Dai.

**Peer review information** *Nature Communications* thanks Trevor Penning
and the other, anonymous, reviewers for their contribution to the peer
review of this work. A peer review file is available.

