## [Peer Review File · Nature Communications]

A three-level regulatory mechanism of the aldo-keto reductase subfamily AKR12DREVIEWER COMMENTS

Reviewer #1 (Remarks to the Author):

This is a potentially important manuscript that describes allosteric regulation of a new member of the aldo-keto reductase (AKR) superfamily, AKRty1 also named as a founder member of the 12D subfamily, AKR12D1. The regulatory mechanism involves (i) high affinity binding of NADPH; (ii) cofactor inhibition at high concentrations of NADPH, where 2'-phosphate of adenosine binds to the catalytic tyrosine Tyr 53 and His 130 to prevent substrate binding; and (iii) binding of the substrate tylosin at an allosteric site distal from the substrate binding site which engages the two additional helices (a7, a10) present in all AKRs that are not part of the (a/b)₈-barrel. Occupancy of the tylosin allosteric site occurs in the absence of cofactor and impinges on the ordered bi bi mechanism that characterizes these enzymes. The authors show that the same 3 regulatory steps may apply across the entire superfamily due to similarity in 3D-structure and the generation of Hill coefficients > 1.0 by performing steady state kinetic analysis. The paper may have a major impact on the AKR field but there are a number of concerns that exist.

1. The authors indicate that the allosteric regulation may be transmitted across monomers of a tetramer or via two tetramers of an octamer in which cofactor binding sites are only partially occupied. However, evidence for these multimeric complexes is only seen in crystal structures and no evidence is provided to support their formation in solution, e.g., ultrafiltration without denaturation or ultracentrifugation. How do we know that the multimers are not due to crystal packing forces? It is noteworthy that when two octamers contain two dormant tetramers and two active tetramers, they belong to a different space group to structures in which one octamer contains two active tetramers suggesting that crystal packing forces may be at play. Furthermore, the crystal structures all likely contain NADP⁺ and not NADPH which would oxidize under crystallographic conditions.

2. There are aspects of the paper that are not novel. The three-step binding model for NADPH and the release of NADP⁺ has been seen before for AKR1C9 (their ref 39), and the same as been observed for the beta-subunits of the K⁺ channel (Tipparaju et al., BBRC (2007) 359: 269). This multistep binding creates a complex mechanism of regulation that needs further examination. If a tetramer forms and each monomer has 3 binding modes for cofactor the factorial could mean that 64 cofactor bound forms are possible and this is without considering the inhibitory binding mode of the cofactor observed at high cofactor concentrations.

3. The authors need to emphasize the novelty of their findings; the 3-step binding model for cofactor binding has been reported before for AKRs; inter-subunit communication has been observed before in the beta-subunit of K⁺ channels; and substrate inhibition has been observed for AKR1D1 but not cofactor inhibition.

4. Is the inhibition by high concentrations of NADPH philologically significant? The K_i observed for this interaction is 148.6 μM where the K_m is 2.98 μM; in addition, the binding sequence proposed predicts a K_d for the tight binding enzyme-NADPH complex of 0.45 μM. Many AKRs have high affinity for NADPH

with K_d values in the sub-micromolar range which reflects the prevailing intracellular concentration of NADPH.

5. The authors propose that high concentrations of NADPH are required to force dormant conformations into active conformations, yet they also propose that high concentrations of NADPH are inhibitory which appears counterintuitive. The authors also claim that when the inhibitory complex results in a non-reactive ternary complex (presumably since there is both productive and non-productive binding of NADPH) this terminates cofactor binding and that the whole process is unique to the AKR superfamily; this seems like a gross overstatement.

6. The authors should place their work in perspective and refer to the classical work of Jeremy Knowles and Gregory Petsko on triose phosphate isomerase (a classical TIM barrel structure).

7. How does the allosteric mechanism play into the ordered bi bi kinetic mechanism of the enzyme and affect the K_{eq} of the reaction which favors carbonyl reduction? Have the authors measured K_{eq} under the different allosteric conditions and reconciled the K_{eq} with a kinetic Haldane?

8. The authors show by using site directed mutagenesis that changes in the allosteric site affects tylosin binding since higher K_m values are observed in the mutants. However, K_m is not affinity in this complex kinetic mechanism. If the authors wish to make an affinity statement, they should conduct single turnover experiments using their stopped flow.

9. The authors make the statement that their allosteric mechanism may be widely applicable to other AKRs. To validate this hypothesis, they conduct steady state kinetics for AKRs that are representative of different subfamilies and report Hill coefficients greater than 1.0. This may be insufficient by itself, when both the cofactor and the carbonyl containing substrate may be allosteric regulators (extended Table 1).

10. The authors are encouraged to inform the curators of the AKR website of their assignment of the new protein to ensure that AKR12D1 has not been assigned to another protein:
<https://akrsuperfamily.org/>

11. The authors imply that at low oxidative stress and high NADPH levels would prevent the reduction of useful aldehydes without identifying what these useful aldehydes might be. This thought requires elaboration. Further this reviewer doubts that the high levels of NADPH needed to achieve cofactor inhibition may not be attainable in cell systems.

12. Can the authors elaborate on the use of high salt in their work. Some of the assays contain 150 mM NaCl which will disrupt the electrostatic linkages required for high affinity cofactor binding. These concentrations were also utilized in their stopped flow experiments to measure tight binding of cofactor.

Minor

-Line 43 "second secret if life" remove meaningless phrase

-Line 63: AKRs do not convert aldehydes to hydroxyl they convert aldehydes to primary alcohols

- Line 81 "is general across three kingdoms of life" rephrase
- Line 163 "nicotinamide head" or "nicotinamide ring"
- Line 291 define "p-carbon"
- Line "710 proteins are from Streptomyces" these are not proteins until expressed they are in silico predictions only

Reviewer #2 (Remarks to the Author):

This manuscript describes a complex allosteric regulatory network in an aldo-keto reductase enzyme. I'm afraid I had trouble making sense of the article or what the key points of broad interest are. A new subfamily of enzymes is found, but its not clear why they are of broad interest. A complex allosteric network is described, but its also unclear what lessons are transferrable to other systems or of broad interest. I'm also unclear about the validity of many of the conclusions. For example, the authors argue for dynamic allostery because there wasn't a clear conformational change in their crystal structure, but there's no follow-up to test this hypothesis. The simulations are also underwhelming.

Reviewer #3 (Remarks to the Author):

This is an interesting manuscript describing the function of an AKR family enzyme, namely the NADPH-dependent reduction of tylosin, and, more impressively, the discovery of multiple levels of catalytic regulation in AKRtyl. Rapid binding kinetics, crystallography, MD simulations are used to uncover the likely molecular basis for substrate inhibition by NADPH, binding activation by NADPH, and allosteric activation by tylosin. Furthermore, the authors used phylogenetic analysis and steady-state kinetics to demonstrate that this three-pronged regulation of catalysis may be widespread within the AKR family. The manuscript could make an important contribution to the AKR field specifically and to the wider field of allosteric control of catalysis, provided the following issues/questions are satisfactorily addressed.

Fig 1c and 1d, Table 1. The k_{cat} 's from varying NADPH and tylosin are too different, when they should be similar. In fact, only one k_{cat} should be reported, as k_{cat} reflects the maximum rate constant for catalysis when the active site is saturated with both NADPH and tylosin. I think the low k_{cat} when tylosin is varied is due to the high level of NADPH used as fixed substrate, which brings the enzyme to a partially inhibited form due substrate inhibition by NADPH. The authors must obtain a saturation curve for tylosin at a lower level of NADPH (I suggest around 25 μ M) to see if the apparent k_{cat} increases, and then report only one k_{cat} (the average of those obtained by varying NADPH and tylosin).

Lines 104-110: In the SI figure 3, there seems to be still some relomysin produced by the negative AKRtyl mutant, when one would expect it there to be none. Is there another enzyme that produces relomysin? Does the aldehyde spontaneously convert to the alcohol at a certain rate, given its high reactivity?

Lines 190-212: It is not clear to me that the observed crystal structures agree with the MWC symmetry model of cooperativity. If different tetramers (which is strange as an allosteric unit since the oligomeric state is octameric in crystal form and in solution) were seen in different conformations that could be attributed to R and T states in the absence of NADP(H) it would make sense to suggest the MWC model. However, since at least one subunit is always bound to NADP(H), how can it be ruled out that the free enzyme (which is not present in the crystal structures) exists in one main conformation and conformational changes were induced by NADPH binding and propagated to the neighbouring subunits, which is the definition of the KNF sequential model? Put another way, what would the two sets of structures need to look like for the KNF model to be considered?

The model in Fig 2e does not represent the MWC model, since no equilibrium of ligand-free protein states is shown.

If the whole octamer is considered, which seems the path of least resistance since that's the oligomeric state in solution, then one can see one tetramer in one conformation (bound to one NADPH molecule) while the other tetramer is in another conformation (without any NADPH). This is not consistent with the MWC model, where all molecules in the oligomer must be in the same state. I cannot find justification for considering the tetramer and not the octamer as the allosteric unit in order to invoke the MWC.

In Extended Data Fig 4 and Supplementary Table 3, k_{obs1} and k_{obs2} are reported from the double-exponential fit. Assuming k_{obs1} is reporting on the cooperative binding of NADPH (k_{obs2} is assumed to be the binding of NADPH to the substrate inhibition site), a replot of k_{obs1} vs [NADPH] concentration could help distinguish between MWC and KNF. I'm surprised the authors did not do that. A hyperbolic decrease in k_{obs} with concentration is diagnostic of MWC (it does not seem to be the case from the table), while a hyperbolic increase is common for KNF, but can also be obtained for MWC under certain conditions, which can be experimentally tested. See, for example, Vogt and Di Cera Conformational Selection or Induced Fit? A Critical Appraisal of the Kinetic Mechanism, Biochemistry 2012 for details.

Lines 341-343: The observation of tylosin binding to form a binary complex with AKRtyl does not rule out an ordered mechanism where NADPH must bind first. The AKRtyl:tylosin binary complex may be a dead-end complex. The kinetic mechanism can only be challenged if this new binary complex is shown to be a part of the reaction conducive to products.

Supplementary Fig 15a, Supplementary Fig 16a, Supplementary Fig 16c 4-nitrobenzaldehyde, fit to the Hill eq seems like a stretch. Please show how poor the fit to the MM eq is.

Line 756: What were the [NADPH]? Since NADPH absorbs at 340 nm, how was the expected inner filtered effect on the Trp fluorescence (presumably what is being measured) controlled for or corrected for? The same experiments titrating the same [NADPH] into Trp alone is the usual way to detect this.

Minor comments:

In Fig 1a, an H^+ is also a substrate, which is not depicted, but should be. Furthermore, the reaction is shown as irreversible. Do the authors know this for a fact? In their HPLC experiments, they had molar

excess NADPH, so it could lead to full conversion.

Can NADP⁺ and relomysin be converted to tylosin and NDAPH (plus H⁺) by AKRtyl?

lines 49 -51: There appears to be a substantial degree of circular logic in the sentence. I suggest the authors rephrase it.

Lines 59-61: surely many non-allosterically regulated proteins are thermostabilised by ligand binding, so the converse cannot be evidence for allosteric regulation.

Also, what are AKR1A1 and AKR1B10? This should be defined/explained for those not versed in this superfamily's nomenclature.

Lines 688-690: add the units for the extinction coefficients.

Lines 694-695: how long was the decrease in abs₃₄₀ monitored for? If more than a few minutes, was evaporation controlled for? This must be stated.

In eqs 1 and 2, the initial rate v must be lowercase. All terms must be defined.

The His-tag does not seem to have been removed from the final proteins. What evidence is there that the presence of the His-tag did not interfere with catalysis and/or allostery in the enzyme used here? If this hasn't yet been done, it must be shown for at least one of the variants that His-tagged and non-His-tagged enzyme forms have the same activity and allosteric regulation.

In the MD simulations, how were the ionization state of ionizable residues, especially histidines, defined at the outset of simulations?

Supplementary Fig 5b: it's not clear that those data could not fit well to the MM equation.

Table 1: Many of the reported precision in the values is undercut by the uncertainty. Please make sure precision is reported only to the decimal place uncertainty allows.

Line 160: replace "close" with "closed"

Extended Dat Fig 6: the arrow should depart from the bond between the pro-R H and C4, not from the H itself. The TS is represented as concerted for hydride and proton transfers. Is that known in this family? If so, cite the reference.

Response to Reviewers' Comments

We sincerely thank the reviewers for the insightful comments and constructive suggestions to improve the quality of our manuscript. We provide a point-by-point response to all comments from reviewers in **BLUE** font below.

Reviewer #1 (Remarks to the Author):

This is a potentially important manuscript that describes allosteric regulation of a new member of the aldo-keto reductase (AKR) superfamily, AKRty1 also named as a founder member of the 12D subfamily, AKR12D1. The regulatory mechanism involves (i) high affinity binding of NADPH; (ii) cofactor inhibition at high concentrations of NADPH, where 2'-phosphate of adenosine binds to the catalytic tyrosine Tyr 53 and His 130 to prevent substrate binding; and (iii) binding of the substrate tylosin at an allosteric site distal from the substrate binding site which engages the two additional helices ($\alpha 7$, $\alpha 10$) present in all AKRs that are not part of the $(\alpha/\beta)_8$ -barrel. Occupancy of the tylosin allosteric site occurs in the absence of cofactor and impinges on the ordered bi bi mechanism that characterizes these enzymes. The authors show that the same 3 regulatory steps may apply across the entire superfamily due to similarity in 3D-structure and the generation of Hill coefficients > 1.0 by performing steady state kinetic analysis. The paper may have a major impact on the AKR field but there are a number of concerns that exist.

Thanks.

1. The authors indicate that the allosteric regulation may be transmitted across monomers of a tetramer or via two tetramers of an octamer in which cofactor binding sites are only partially occupied. However, evidence for these multimeric complexes is only seen in crystal structures and no evidence is provided to support their formation in solution, e.g., ultrafiltration without denaturation or ultracentrifugation. How do we

know that the multimers are not due to crystal packing forces? It is noteworthy that when two octamers contain two dormant tetramers and two active tetramers, they belong to a different space group to structures in which one octamer contains two active tetramers suggesting that crystal packing forces may be at play. Furthermore, the crystal structures all likely contain NADP⁺ and not NADPH which would oxidize under crystallographic conditions.

We thank the reviewer for the important questions and suggestions.

- i. We surmised that AKRtyl behaves as an octamer in the solution state primarily by size-exclusion chromatography (Supplementary Fig. 4). However, as the reviewer was concerned, the accuracy of size-exclusion chromatography is relatively low which cannot give conclusive evidence about the authentic oligomerization state, we further performed size-exclusion chromatography coupled with multi-angle light scattering (SEC-MALS) experiment to accurately determine the oligomerization state of AKRtyl in solution (Figure R1). The molecular weight of an AKRtyl monomer is 38.4 kDa, and the molecular weight determined in the SEC-MALS experiment is 285 kDa which is about 7.5 times of a monomer, so we conclude that AKRtyl behaves as an octamer in solution.

Thanks for the reviewer's suggestion. We have integrated the SEC-MALS result into the manuscript (line 154-155) and shown as Supplementary Fig. 5.

Figure R1. Determination of the molecular weight of AKRtyl in solution by size-

exclusion chromatography coupled with multi-angle light scattering (SEC-MALS).

- ii. Although the crystal structure containing two octamers in an asymmetric unit has different space group and cell parameters compared with the crystal structure containing one octamer, their octamer configurations are highly consistent (Figure R2a-c). In addition, by careful analysis of the crystal packing containing active (R) and dormant (T) conformations (Figure R2d-k), we find that the region (shown as transparent pink ellipse) used to distinguish between R and T in all the subunits was not occupied or squeezed by the neighboring asymmetric units (shown as cartoon colored in grey-white). We then reposition the subunit with R conformation by aligning it to the subunit with T conformation (Figure R2m), as well as repositioning the subunit with T conformation with the subunit with R conformation (Figure R2o), and we find that both the R and T conformations are well accommodated in their new positions and do not conflict with other asymmetric units at the junction (Figure R2l and n). We therefore speculate that the different conformational states should not result from crystal packing.

Thanks for the reviewer's question. We have integrated the analysis of crystal packing in the manuscript (line 195-196) and shown as Supplementary Fig. 10.

Figure R2. Crystal packing analysis of the two purified-state AKRtyl structures. **a-c**, One-to-one alignment of three octamers in the two purified-state structures. **d-g**, Interactions of each subunit of the dormant conformational tetramer with neighboring asymmetric units. **h-k**, Interactions of each subunit of the active conformational tetramer with neighboring asymmetric units. The dormant conformation is colored green, the active conformation is colored blue, the neighboring asymmetric units are colored grey-white and the structural regions used to distinguish between R and T are shown as transparent ellipse colored in pink. **l-o**, Alignment of subunits with one conformation to another conformation (**m**, R align to T; **n**, T align to R). The junction with neighboring asymmetric units is shown as rectangle colored in yellow.

iii. To test whether NADP^+ or NADPH was present in the crystal structure, we recrystallized the AKRtyl-NADPH complex and measured full UV-vis spectra in different stages. When the crystals appeared, we first measured the UV-vis spectra of the whole droplet, then we picked up some crystals, washed them with the crystallization pool solution, dissolved them in the protein storage buffer, and measured the UV-vis spectra again. Compared with the initial solution before crystallization, there was no absorption at 340 nm in either the whole droplet or

the crystals (Figure R3). Thus, it is indeed NADP^+ that should present in the crystals, as pointed out by the reviewer.

Thanks for the reviewer's reminder. We have added cofactor analysis in the manuscript (line 161-163) and shown as Supplementary Fig. 8.

Figure R3. UV-vis spectra measurements of AKRtyl-NADPH complex crystals.

2. There are aspects of the paper that are not novel. The three-step binding model for NADPH and the release of NADP^+ has been seen before for AKR1C9 (their ref 39), and the same as been observed for the beta-subunits of the K^+ channel (Tipparaju et al., BBRC (2007) 359: 269). This multistep binding creates a complex mechanism of regulation that needs further examination. If a tetramer forms and each monomer has 3 binding modes for cofactor the factorial could mean that 64 cofactor bound forms are possible and this is without considering the inhibitory binding mode of the cofactor observed at high cofactor concentrations.

We thank the reviewer for this important reminder. There have been reports of the three-step binding model for cofactor NADPH in the AKR superfamily. However, as elucidated below, AKRtyl is different and unique.

In the case of AKR1C9, the authors found through fluorescence quenching experiments that the quenching traces was better fitted with a double-exponential decay equation. Combining the structural information, the authors proposed a three-step binding model (Figure R4a). The model involves the formation of the initial loose

complex AKR1C9·NADPH, followed by two subsequent conformational changes that result in a tight binding complex. The first conformational change may be attributed to the formation of an electrostatic linkage between Arg276 and the 2'-phosphate of AMP and yields AKR1C9*·NADPH complex. The second conformational change yields the tight AKR1C9**·NADPH complex. These two steps of conformational change involve fluorescence quenching of Trp86, Trp224 and are therefore detectable (Figure R5a-b). The loop where Trp224 is located is essential for the formation of the tight complex, so the authors were able to detect the conformational change for this step (Figure R5a-b).

We also propose a three-step model for NADPH binding to AKRtyl (Figure R4b), and it is distinct from that of AKR1C9 as evidenced by their structural difference.

- i. The first step is the NADP(H)-induced conversion from the dormant to the active conformation. This step was observed both from crystal structure and the fluorescence quenching assays of NADPH binding to AKRtyl in the equilibrium solution state (Figure 4b-d in the main text).
- ii. The second step was observed based on structural information and transient fluorescence quenching. We found that the quenching traces were better fitted with a double-exponential decay equation, akin to AKR1C9. However, as shown in Figure R5a-b, the binding pocket of cofactor NADP(H) does not contain equivalent tryptophan (Trp86 and Trp224 in AKR1C9), but only Trp214 (corresponding to Tyr216 in AKR1C9) which lies below the cofactor and NADP(H) binding can cause fluorescence quenching on it. On the other hand, the structure shows that AKRtyl undergoes only one conformational change upon binding the cofactor, which is a long cofactor binding loop that wraps around the cofactor to form a tight complex (Figure R5c). Thus, we were able to detect the step of the loose complex AKRtyl*·NADPH^c formation by fluorescence quenching experiment, but not the conformational change for forming the tight complex (AKRtyl**·NADPH^c) because there is no tryptophan on the loop. We therefore combined the entire process of NADP(H) binding, from forming the loose complex to the one-step conformational change for tight complex formation, into the second step of the three-step model (Figure R4b), as only one decay signal corresponding to this

process is observed in transient fluorescence quenching experiments.

- iii. The second decay signal in the double-exponential decay observed in transient fluorescence quenching experiments is caused by the binding of inhibitory NADPH. This is the third step of our three-step model (Figure R4b). This step was proposed on the basis of crystal structures, transient fluorescence quenching experiments and fluorescence quenching experiments in the equilibrium solution state. Inhibitory NADPH binds to the substrate pocket, causing Trp331 fluorescence quenching. To further validate the relationship of Trp331 between the second decay signal in the transient fluorescence quenching, we measured the titration of NADPH against W331A using stopped-flow. We find that the fluorescence quenching traces fit the mono-exponential decay equation very well, while the double-exponential decay equation can only be fitted at high NADPH concentrations, but the quenching amplitude and quenching rate of the second process are weak (Figure R6 and Table R1). These results further confirm that inhibitory NADP(H) binding to the substrate pocket is the second decay signal observed in the transient fluorescence quenching, which is different from that observed in AKR1C9.

To summarize, our proposed three-step model of NADPH binding to AKRtyl is different from that of AKR1C9. The three steps include the dormant to active conformational transition induced by low concentration of NADPH, the productive cofactor binding at the active site, and the binding of inhibitory NADPH. All these steps are taken into account in an integrated manner.

We have integrated the three-step NADPH binding model analysis into the manuscript (line 288-292, 338-342), Supplementary Fig. 14, Supplementary Fig. 16 and Supplementary Table 5.

Figure R4. Three-step model for NADPH binding to AKR1C9 (a) and AKRtyl (b).

Figure R5. Binding of NADP(H) to AKRtyl and AKR1C9. **a-b**, Structural comparison of NADP(H) binding to AKRtyl and AKR1C9. **c**, Conformational change of AKRtyl upon binding of NADP(H). **d**, AKRtyl binds inhibitory NADPH.

Figure R6. Time course of representative binding of NADPH (25 μM) to AKRtyl-W331A (0.2 μM). **a**, Fitting with the double-exponential decay equation. **b**, Fitting with the mono-exponential decay equation.

Table R1. Fitting of the fluorescent quenching curves of NADPH to AKRtyl-W331A with either double or single exponential model.

NADPH (μM)	Model	$(\Delta F)_1$ (V)	k_{obs1} (s^{-1})	$(\Delta F)_2$ (V)	k_{obs2} (s^{-1})	R^2	
100	Double exponential	1.555 ± 0.01	102.04 ± 1.45	0.09 ± 0.01	10.95 ± 3.19	0.9914	Amplitude and rate of the 2 nd process are weak
	Single exponential	1.58 ± 0.01	91.75 ± 0.76	-	-	0.9933	
50	Double exponential	1.41 ± 0.03	79.33 ± 1.70	0.12 ± 0.03	18.62 ± 4.05	0.9963	Poor fit
	Single exponential	1.49 ± 0.01	69.56 ± 0.48	-	-	0.9954	
25	Double exponential	1.35 ± 0.01	58.74 ± 0.81	0.38 ± 4.81	0.51 ± 6.94	0.9970	Poor fit
	Single exponential	1.36 ± 0.01	56.08 ± 0.34	-	-	0.9966	
10	Double exponential			unfit			
	Single exponential	0.90 ± 0.01	34.17 ± 0.29	-	-	0.9942	
5	Double exponential			unfit			
	Single exponential	0.57 ± 0.01	27.94 ± 0.36	-	-	0.9875	
2.5	Double exponential			unfit			
	Single exponential	0.33 ± 0.01	23.76 ± 0.53	-	-	0.9677	

3. The authors need to emphasize the novelty of their findings; the 3-step binding model for cofactor binding has been reported before for AKRs; inter-subunit communication has been observed before in the beta-subunit of K⁺ channels; and substrate inhibition has been observed for AKR1D1 but not cofactor inhibition.

We appreciate the important suggestion. The novelty of our study is mainly reflected in the three-level structural regulation present in AKRtyl. We have elucidated the molecular mechanisms of this three-level regulation from multiple perspectives, including structure and kinetics, where the cofactor NADPH has two levels of regulation on AKRtyl and substrate tylosin has one level.

The two-level regulation for NADPH are (1) allosteric regulation of NADPH at micromolar level concentrations, inducing a shift in AKRtyl from a dormant to an active conformation which employs the MWC allosteric model, and (2) cofactor inhibition at submillimolar concentrations by “robbing” the substrate pocket. This differential response to different concentrations of the same ligand is special and novel. The three-step binding model we propose for the NADPH binding in AKRtyl is also unique in the AKR family.

In addition, binding of substrate tylosin at a remote site results in positive allosteric effect on enzyme catalysis, which induces loop movement and opening of active pockets. Whilst the effect of ligand on enzyme stability has been reported in other AKRs and it was speculated that there may be allosteric regulation, our work provides direct structural evidence.

4. Is the inhibition by high concentrations of NADPH philologically significant? The K_i observed for this interaction is 148.6 μM where the K_m is 2.98 μM ; in addition, the binding sequence proposed predicts a K_d for the tight binding enzyme-NADPH complex of 0.45 μM . Many AKRs have high affinity for NADPH with K_d values in the sub-micromolar range which reflects the prevailing intracellular concentration of NADPH.

We appreciate the important question. We think that whether the inhibition of

AKRtyl by NADPH is physiologically significant depends on whether the intracellular NADPH concentration can reach the K_i value (148.6 μM). By reviewing the literatures, we found that there are several studies that have measured the concentration of NADPH in cells. The intracellular NADPH concentration reaches 120 μM in *Escherichia coli* K-12 NCM3722¹, 190 μM in *Corynebacterium glutamicum* ATCC13032² and over 300 μM in *Synechococcus* PCC7942³. In addition, we also reviewed information on intracellular NADPH concentrations in *Streptomyces*. The intracellular NADPH concentration is $1.89 \pm 0.05 \mu\text{mol/g}$ biomass in *Streptomyces tsukubaensis* NRRL 18488⁴, approximating the conversion of 1 g of biomass to 1 mL gives a NADPH concentration of $1.89 \pm 0.05 \text{ mM}$. We found that most organisms have intracellular NADPH concentrations in the sub-millimolar range. Therefore, we think that high concentration NADPH inhibition is physiologically significant.

We have integrated these findings from literature in the manuscript (line 129-131).

5. The authors propose that high concentrations of NADPH are required to force dormant conformations into active conformations, yet they also propose that high concentrations of NADPH are inhibitory which appears counterintuitive. The authors also claim that when the inhibitory complex results in a non-reactive ternary complex (presumably since there is both productive and non-productive binding of NADPH) this terminates cofactor binding and that the whole process is unique to the AKR superfamily; this seems like a gross overstatement.

Thanks to the reviewer for the question. The concentration of NADPH required to force dormant conformations into active conformations are at the micromolar range, while the inhibitory effect occurs at higher concentrations in the sub-millimolar range. We apologize that we did not state it clearly.

In the case of AKRtyl, productive NADPH binding, consistent with other reported AKRs, binds to the conventional cofactor binding site and is necessary for its catalytic function. While non-productive NADPH binding robs the substrate binding pocket and prevents substrate loading, thereby generating the non-reactive complexes. This feature makes AKRtyl special in the AKR superfamily.

We thank the reviewer for pointing this out, and we have removed inappropriate sentences.

6. The authors should place their work in perspective and refer to the classical work of Jeremy Knowles and Gregory Petsko on triose phosphate isomerase (a classical TIM barrel structure).

We thank the reviewer for this suggestion. The TIM barrel, also known as an alpha/beta barrel, represents a compact but adaptable scaffolding which is ubiquitous, with approximately 10% enzymes adopting this fold.⁵ TIM barrel is named after triose-phosphate isomerase, a conserved metabolic enzyme.⁶ Jeremy Knowles and Gregory Petsko have done a lot of pioneering work on the catalytic mechanism^{7, 8}, structure⁹, and evolution¹⁰ of triose-phosphate isomerase, which has provided an excellent foundation and made outstanding contributions to the field of enzyme catalysis.

We have referred to the classical work of Jeremy Knowles and Gregory Petsko on triose phosphate isomerase in the manuscript (line 152-153).

7. How does the allosteric mechanism play into the ordered bi-bi kinetic mechanism of the enzyme and affect the K_{eq} of the reaction which favors carbonyl reduction? Have the authors measured K_{eq} under the different allosteric conditions and reconciled the K_{eq} with a kinetic Haldane?

We appreciate the reviewer for this important suggestion and question. To determine the effect of the ligands on the equilibrium constant of the reaction, we varied the concentration of NADPH or tylosin and fixed the concentration of the other to study the effect of a single factor. We found that the equilibrium constant K_{eq} exhibits an increase and then a decrease as a function of NADPH concentration (as shown in Table R2), e.g., $K_{eq} = 0.86$ when $[NADPH] = 5 \mu\text{M}$, and $K_{eq} = 81.37$ when $[NADPH] = 50 \mu\text{M}$, an almost 100-fold (2 orders of magnitude) increase. This means that NADPH pushes the equilibrium towards reduction within a certain concentration range. And when $[NADPH] = 200 \mu\text{M}$, $K_{eq} = 0.89$, which is nearly 100 times (2 orders of magnitude) lower compared to $[NADPH] = 50 \mu\text{M}$. This means that by further increasing the

NADPH concentration, the equilibrium is pulled back from the reduction. And this is similar to our observation of a microscopic molecular mechanism whereby at micromolar level concentrations, NADPH induces a conformational shift from dormant to active conformations, favoring the reaction, whereas at sub-millimolar concentrations, NADPH acts as an inhibitor binding to the substrate pocket and inhibiting of the reaction.

The effect of tylosin on the reaction equilibrium was different, as K_{eq} increased almost 20-fold as the tylosin concentration was increased from the micromolar [25 μM] to the submillimolar level [100-500 μM], whereas a further increase in tylosin concentration [1 mM] resulted in a smaller change in K_{eq} , which was in the same order of magnitude (as Table R3 shown). This implies that tylosin is able to pull the equilibrium toward reduction. This is also consistent with our observation of a microscopic molecular mechanism whereby tylosin acts as an allosteric modulator to facilitate catalysis.

Next, we would like to reconcile the K_{eq} with a kinetic Haldane as the reviewer suggested. Therefore, it is necessary to determine the kinetics of the reverse reaction – the oxidation of relomycin. We reduced tylosin by AKRtyl and purified it by preparative HPLC to obtain relomycin as it is not easy to purchase. We found that AKRtyl can use the cofactor NADP^+ to oxidize relomycin, but the oxidation rate is very low, for example, the oxidation rate under 200 μM NADP^+ and 1 mM relomycin is 1.98×10^{-4} $\mu\text{M/s}$, while the reduction rate under 200 μM NADPH and 1 mM tylosin is 0.38 μM , which is almost 2000-fold higher than the oxidation rate. When we measured the kinetics of the oxidation reaction, we found that perhaps due to the low affinity of AKRtyl for relomycin, we used up the purified relomycin but did not reach saturation in the kinetic curve (Figure R7). Therefore, we are unable to obtain complete oxidation reaction kinetics and cannot perform a kinetic Haldane reconciliation.

Thanks for the reviewer's suggestions. We have integrated these results related to K_{eq} into the manuscript (line 310-329, line 400-409) and Extended Data Table 2.

Figure R7. The oxidation rate of relomycin catalyzed by AKRtyl ($[\text{NADP}^+] = 200 \mu\text{M}$).

Table R2 Equilibrium constants at different NADPH concentrations. ^a

NADPH (μM)	5	10	25	50	100	200
Keq	0.86 ± 0.04	20.76 ± 8.73	44.02 ± 5.88	81.37 ± 2.38	10.50 ± 1.56	0.89 ± 0.03

^a The concentration of tylosin in the reaction system was fixed at $50 \mu\text{M}$.

Table R3 Equilibrium constants at different tylosin concentrations. ^a

Tylosin (μM)	25	50	100	250	500	1000
Keq	2.18 ± 1.31	11.70 ± 3.46	47.26 ± 9.14	33.77 ± 2.13	51.45 ± 12.10	25.16 ± 1.48

^a The concentration of NADPH in the reaction system was fixed at $100 \mu\text{M}$.

8. The authors show by using site directed mutagenesis that changes in the allosteric site affects tylosin binding since higher K_m values are observed in the mutants. However, K_m is not affinity in this complex kinetic mechanism. If the authors wish to make an affinity statement, they should conduct single turnover experiments using their stopped flow.

We thank the reviewer for pointing out the question and suggestion. We performed single turnover experiments to determine the K_d values of WT, E193A, R193W, and R257W for tylosin using stopped-flow. The enzyme concentration was $0.5 \mu\text{M}$, the NADPH concentration was $0.3 \mu\text{M}$, and the tylosin concentration varied from 0 to 10

mM. Rate constant k_{obs} values were obtained by fitting traces to the mono-exponential equation. We found that the single turnover kinetic data were better fitted with the Hill equation with a K_d value of 1.96 mM, which agrees with the allosteric effect of tylosin (Figure R8a). Mutants of tylosin allosteric binding site all have much higher K_d values for tylosin (Figure R8b-c) than that of WT, suggesting that disruption of the allosteric site reduces the affinity of AKRtyl for tylosin.

Thanks for the reviewer's suggestions. We have integrated these results related to K_d measured by single turnover experiments into the manuscript (line 396-398) and Supplementary Fig. 22.

Figure R8. Single turnover kinetic curves of AKRtyl WT (a) and substrate allosteric binding site mutants (b-d). The data were fitted with both the hyperbolic equation (blue line) and the allosteric-sigmoidal Hill equation (red line).

9. The authors make the statement that their allosteric mechanism may be widely applicable to other AKRs. To validate this hypothesis, they conduct steady state kinetics

for AKRs that are representative of different subfamilies and report Hill coefficients greater than 1.0. This may be insufficient by itself, when both the cofactor and the carbonyl containing substrate may be allosteric regulators (extended Table 1).

We thank the reviewer for the question. In these kinetic assays we used a single factor change strategy, fixing the concentration of NADPH and varying only the concentration of the substrate. Thus, the effect of NADPH was fixed and the positive synergistic effect seen in the kinetics was due to the change in substrate concentration alone.

10. The authors are encouraged to inform the curators of the AKR website of their assignment of the new protein to ensure that AKR12D1 has not been assigned to another protein: <https://akrsuperfamily.org/>

We thank the reviewer for this important reminder. The AKRs superfamily database contains more than 200 AKRs from 17 families, and almost every AKR has a detailed functional note. In addition to this, the curators developed a systematic nomenclature which greatly facilitate our understanding of AKRs. With the reviewer's reminder, we found that AKR12D1 was indeed assigned to another protein. However, after sequence analysis, we found that the AKR12D1 in the database should be renamed as AKR16B1. After contacting the curators, AKRtyl was finally named as AKR12D1 and the original AKR12D1 was renamed as AKR16B1. Thanks again to the reviewer for the reminder.

11. The authors imply that at low oxidative stress and high NADPH levels would prevent the reduction of useful aldehydes without identifying what these useful aldehydes might be. This thought requires elaboration. Further this reviewer doubts that the high levels of NADPH needed to achieve cofactor inhibition may not be attainable in cell systems.

We appreciate the important questions regarding the physiological significance. Useful intracellular aldehydes include glyceraldehyde 3-phosphate, an important intermediate in metabolic pathways such as glycolysis, and erythrose 4-phosphate, an important intermediate in metabolic pathways such as the pentose phosphate pathway.

We found that AKRtyl was able to reduce glyceraldehyde 3-phosphate and erythrose 4-phosphate with catalytic activities of $15.97 \pm 1.37 \mu\text{M/s/mg}$ and $56.16 \pm 4.68 \mu\text{M/s/mg}$, respectively. In addition, as mentioned above, most organisms have intracellular NADPH concentrations in the sub-millimolar range where inhibition can be achieved. Therefore, we imply that at low oxidative stress and high NADPH levels would prevent the reduction of useful aldehydes by AKRtyl.

We have integrated these results related to useful aldehydes into the manuscript (line 139-141, line 565-567) and Supplementary Table. 1.

12. Can the authors elaborate on the use of high salt in their work. Some of the assays contain 150 mM NaCl which will disrupt the electrostatic linkages required for high affinity cofactor binding. These concentrations were also utilized in their stopped flow experiments to measure tight binding of cofactor.

The assay buffer contained 150 mM NaCl. The reason for using such high salt is as follows: first, 150 mM NaCl is close to the physiological salt concentration (0.9% NaCl);¹¹ second, the protein is relatively stable at this salt concentration, as it easily precipitated at lower salt concentrations.

We reduced the NaCl concentration in our assays to 10 mM and found that although at lower salt concentration there is a stronger signal for fluorescence quenching (Figure R9), the curve (after subtracting the tryptophan control as suggested by the reviewer 3) was still better fitted with the double-exponential decay equation than with the mono-exponential decay equation (as Figure R10 and Table R4 shown). These results are in agreement with our previous observations (Extended Data Fig 4 and Supplementary Table 4).

Figure R9. Representative fluorescence quenching traces of AKRtyl (0.2 μM) binding to NADPH (25 μM) with the buffer containing 150 mM NaCl (red line) and 10 mM NaCl (blue line).

Figure R10. Representative binding trace of AKRtyl (0.2 μM) to NADPH (25 μM) (with the buffer containing 10 mM NaCl) after subtracting the tryptophan control and fitting with double-exponential decay (red line) and mono-exponential decay equations (blue line).

Table R4. Fit of the real time fluorescent quenching of NADPH to AKRtyl with two models (the buffer containing 10 mM NaCl).

NADPH (μM)	Model	$(\Delta F)_1$ (V)	k_{obs1} NADPH ^c Step2 (s ⁻¹) ^a	$(\Delta F)_2$ (V)	k_{obs1} 2 st NADPH ⁱ Step3 (s ⁻¹) ^b	R^2
100	Double exponential	1.13 \pm 0.15	182.30 \pm 15.98	0.93 \pm 0.16	73.95 \pm 5.21	0.9914
	Single exponential	1.94 \pm 0.01	109.89 \pm 0.86	-	-	0.9881
50	Double exponential	1.40 \pm 0.07	181.00 \pm 8.49	1.13 \pm 0.08	61.80 \pm 2.25	0.9951
	Single exponential	2.34 \pm 0.01	97.63 \pm 0.67	-	-	0.9909
25	Double exponential	1.78 \pm 0.06	153.39 \pm 4.80	1.10 \pm 0.07	50.29 \pm 1.61	0.9968
	Single exponential	2.66 \pm 0.01	87.81 \pm 0.57	-	-	0.9817
10	Double exponential	1.88 \pm 0.06	124.36 \pm 3.22	1.08 \pm 0.06	40.36 \pm 1.21	0.9977
	Single exponential	2.74 \pm 0.01	73.11 \pm 0.45	-	-	0.9927
5	Double exponential	1.29 \pm 0.07	115.22 \pm 4.66	1.28 \pm 0.07	39.95 \pm 1.12	0.9974
	Single exponential	2.38 \pm 0.01	60.31 \pm 0.34	-	-	0.9938
2.5	Double exponential	0.98 \pm 0.11	82.83 \pm 6.17	0.48 \pm 0.12	32.40 \pm 3.83	0.9890
	Single exponential	1.38 \pm 0.07	57.08 \pm 0.48	-	-	0.9869

^a The binding process of the NADPH^c corresponds to the step2 of the three-step model.

^b The binding process of the NADPHⁱ corresponds to the step3 of the three-step model.

Minor

-Line 43 “second secret if life” remove meaningless phrase

We have removed the meaningless phrase as suggested. (line 41-42)

-Line 63: AKRs do not convert aldehydes to hydroxyl they convert aldehydes to primary alcohols

We have corrected the “hydroxyl” to “primary alcohols” as suggested. (line 53, 67)

-Line 81 “is general across three kingdoms of life” rephrase

We have rephrased the sentence: “the substrate cooperativity is a universal mechanism in AKRs across three kingdoms of life”. (line 84-85)

-Line 163 “nicotinamide head” or “nicotinamide ring”

We have changed the “the ring structure of nicotinamide head” to “the nicotinamide ring” as suggested. (line 176)

-Line 291 define “*p*-carbon”

We have corrected the “*p*-carbon” to “*para*-carbon” as suggested. (line 360-361)

-Line 393 “710 proteins are from *Streptomyces*” these are not proteins until expressed they are in silico predictions only

We have corrected the “710 proteins are from *Streptomyces*” to “710 sequences are from *Streptomyces*” as suggested. (line 470)

Reviewer #2 (Remarks to the Author):

This manuscript describes a complex allosteric regulatory network in an aldo-keto reductase enzyme. I'm afraid I had trouble making sense of the article or what the key points of broad interest are. A new subfamily of enzymes is found, but its not clear why they are of broad interest. A complex allosteric network is described, but its also unclear what lessons are transferrable to other systems or of broad interest. I'm also unclear about the validity of many of the conclusions. For example, the authors argue for dynamic allostery because there wasn't a clear conformational change in their crystal structure, but there's no follow-up to test this hypothesis. The simulations are also underwhelming.

We thank the reviewer for the questions. Aldo-keto reductases (AKRs) are a large superfamily of redox classes distributed in nearly all phyla. AKRs can metabolize a wide range of substrates, including drugs, carcinogens, and reactive aldehydes, many of which are harmful to organisms. Thus, AKRs play a central role in the metabolism of these agents, and this can lead to their detoxication.¹² For example, enzymes such as human AKR1A1 and AKR1B10, have been shown to be closely associated with many human diseases due to their important functions in the metabolism of toxic aldehydes and ketones.^{13,14} The enzyme in our research, AKRtyl, represents a new AKR subfamily that is widely distributed in *Streptomyces*, which are known as antibiotic production factories. We speculate that this type of AKRs also function primarily as detoxifiers in *Streptomyces*. Therefore, elucidating the molecular mechanisms will help us to understand the *Streptomyces* physiology. Also, AKRtyl reduces the useful antibiotic tylosin to the byproduct relomycin, so studies on AKRtyl can also help guide the production of tylosin.

Our study reveals the three-level regulatory mechanism of AKRtyl from multiple perspectives, including kinetics and structures, which is novel and unusual. Especially, AKRtyl responds differently to different concentrations of the same ligand, NADPH. In addition, studies of allosteric regulation in the AKR superfamily are rare, with one allosteric example lacking stronger evidence such as x-ray structures.¹⁵ Our studies

confirm the allosteric regulation of AKRtyl by the substrate tylosin. Although there are no obvious conformational changes induced by the allosteric substrate, molecular dynamics simulations show the opening of the active pocket under allostery by inducing loop movement near the pocket. We hypothesize that this three-level regulatory strategy of AKRtyl will help cope with intracellular redox levels, to better perform detoxification functions and to reduce the consumption of useful aldehydes *in vivo*. Our studies will be of great interest in the AKR field, and more broadly, in the field of protein allostery.

Reviewer #3 (Remarks to the Author):

This is an interesting manuscript describing the function of an AKR family enzyme, namely the NADPH-dependent reduction of tylosin, and, more impressively, the discovery of multiple levels of catalytic regulation in AKRtyl. Rapid binding kinetics, crystallography, MD simulations are used to uncover the likely molecular basis for substrate inhibition by NADPH, binding activation by NADPH, and allosteric activation by tylosin. Furthermore, the authors used phylogenetic analysis and steady-state kinetics to demonstrate that this three-pronged regulation of catalysis may be widespread within the AKR family. The manuscript could make an important contribution to the AKR field specifically and to the wider field of allosteric control of catalysis, provided the following issues/questions are satisfactorily addressed.

Thanks.

Fig 1c and 1d, Table 1. The k_{cat} 's from varying NADPH and tylosin are too different, when they should be similar. In fact, only one k_{cat} should be reported, as k_{cat} reflects the maximum rate constant for catalysis when the active site is saturated with both NADPH and tylosin. I think the low k_{cat} when tylosin is varied is due to the high level of NADPH used as fixed substrate, which brings the enzyme to a partially inhibited form due substrate inhibition by NADPH. The authors must obtain a saturation curve for tylosin at a lower level of NADPH (I suggest around 25 μM) to see if the apparent k_{cat} increases, and then report only one k_{cat} (the average of those obtained by varying NADPH and tylosin).

We thank the reviewer for this suggestion. We have re-determined the kinetics of AKRtyl on tylosin at 25 μM NADPH and 200 μM NADPH. We found a significant increase of the k_{cat} value at 25 μM NADPH (Figure R11 and Table R5) and this k_{cat} value (4.65 s^{-1}) is highly consistent with the previously measured k_{cat} value of 4.38 s^{-1} for the NADPH kinetic (Table 1 in the main text).

We have integrated these results into the manuscript as shown in Table 1 and Figure 1d.

Figure R11. Kinetic curves of AKRtyl on tylosin at 25 μM NADPH (red line) and 200 μM NADPH (blue line). These kinetic data were fitted with the Hill equation.

Table R5. Kinetic values of AKRtyl on tylosin at 25 μM NADP and 200 μM NADPH.

NADPH (μM)	$K_{0.5}$ (μM)	k_{cat} (s^{-1})	k_{cat}/K_m ($\text{s}^{-1} \mu\text{M}^{-1}$)	Hill coeff n_H ^a	
25	278.00 ± 12.32	4.65 ± 0.09	16.72×10^{-3}	1.76 ± 0.11	Pos. coop.
200	242.30 ± 12.27	1.52 ± 0.03	6.27×10^{-3}	1.79 ± 0.13	Pos. coop.

^a The Hill coefficient n_H of substrate binding as a measure of cooperativity ($n_H > 1$, positive cooperativity; $n_H = 1$, no cooperativity; $n_H < 1$, negative cooperativity).

Lines 104-110: In the SI figure 3, there seems to be still some relomysin produced by the negative AKRtyl mutant, when one would expect it there to be none. Is there another enzyme that produces relomysin? Does the aldehyde spontaneously convert to the alcohol at a certain rate, given its high reactivity?

In the biosynthesis of tylosin, the generation of C20 aldehyde group is catalysed by sequential oxidation of a cytochrome P450 enzyme TyII (as shown in Figure R12). The remaining relomycin is mainly the result of the post-modification product of TyII-catalyzed hydroxyl intermediate form, 20-dihydro-5-O-mycaminosyl-tylactone. In our recently published study, overexpression of TyII was found to further eliminate relomycin.¹⁶

Figure R12. Cytochrome P450 TyII catalyses the biosynthesis of tylosin C20 aldehyde group.

Lines 190-212: It is not clear to me that the observed crystal structures agree with the MWC symmetry model of cooperativity. If different tetramers (which is strange as an allosteric unit since the oligomeric state is octameric in crystal form and in solution) were seen in different conformations that could be attributed to R and T states in the absence of NADP(H) it would make sense to suggest the MWC model. However, since at least one subunit is always bound to NADP(H), how can it be ruled out that the free enzyme (which is not present in the crystal structures) exists in one main conformation and conformational changes were induced by NADPH binding and propagated to the neighboring subunits, which is the definition of the KNF sequential model? Put another way, what would the two sets of structures need to look like for the KNF model to be considered?

Thanks to the reviewer for the suggestions and questions. As the reviewer points out, it would make sense to propose the MWC model if different tetramers were seen in different conformations that could be attributed to R and T states in the absence of NADP(H). To obtain the apo structure of AKRtyl, we used a protein denaturation buffer (10 mM HEPES, 300 mM NaCl, 2 M KBr, 2 M CO(NH₂)₂) to remove the carried cofactors, then obtained cofactor-free proteins after refolding on the Ni-NTA column. Since none of the previous crystallization conditions produced crystals, we re-screened the crystallization conditions for the apo-protein, and after diffraction we succeeded to

obtain two sets of apo structures without any cofactors. The first set of apo structure is similar to our previous second purified-state structure, with two octamers in an asymmetric unit and one active (R_4) tetramer and one dormant (T_4) tetramer per octamer (R_4T_4). The second set of apo structure has only two subunits in an asymmetric unit, and both subunits are in the active conformation (R); by combining with other asymmetric units, we can obtain an octamer containing two active tetramers (R_4R_4). We didn't get the apo structures in the T_4T_4 state and speculate that structures in this state may not crystallize due to energetic issues. Given that cofactor-independent R- and T-tetramers are observed in the apo structures, we think that the tetramer is in equilibrium between the T and R states. In addition, the structure of the tetramer is always symmetrical under different ligand states (as FigureR13 shown). All of these features fit well with the MWC model, so it is reasonable for us to propose a tetramer-based MWC model for AKRtyl.

However, as the reviewers wondered, AKRtyl is actually an octamer, and it is odd to have a tetramer as an allosteric unit. If the tetramer is treated as a monomer, on the scale of the octamer (treated as a dimer), it makes sense to take the KNF sequential model, since asymmetric structures have been observed. Yet it fails to explain the symmetry within the tetramer because the four subunits in the tetramer always exhibit concerted structural features. If the overall KNF model is to be considered, then in the structure we should have observed asymmetries within the tetramer, e.g. subunits that bind NADP(H) exhibit R conformations, while those that do not bind NADP(H) exhibit T conformations.

Indeed, similar examples exist in the literature such as the chaperonin GroEL, which structurally behaves as a tetradecamer composed of two heptameric rings.¹⁷ GroEL has two levels of allostery: one within each ring and the second between rings. In the first level, each heptameric ring is in equilibrium between the T and R states, in accordance with the MWC model of cooperativity. A second level of allostery is between the rings which undergoes sequential KNF-type transitions from the T_7T_7 state via the T_7R_7 state to the R_7R_7 state. The researchers thus proposed Nested Cooperativity for GroEL.^{18, 19} AKRtyl, much like GroEL, exhibits MWC-type concerted transitions

from T₄ state to R₄ state within the tetrameric ring. We surmise that there are also transitions from the T₄T₄ state via the T₄R₄ state to the R₄R₄ state, but we can't guarantee that there is an allostery in the transitions, as there is no evidence that a conformational shift in one tetramer would have an effect on the other. We can only confirm the concerted model within the tetrameric ring, where NADP(H) binding drives the equilibrium towards the R conformation.

Thanks for the reviewer's suggestions. We have integrated these results into the manuscript (line 217-236), Table 2 and Extended Data Fig. 3.

Figure R13. The active (R) and dormant (T) conformations with tetramer as the

allosteric unit in AKRtyl. Cartoon (top) and conformational pattern (bottom) of AKRtyl structures in 6 different conformations and different ligand states. Active subunits (R) are colored light blue, NADPH bound dark blue, dormant subunits (T) are colored green. **a**, the first apo structure (R₄T₄ octamer). **b**, the second apo structure (R₄R₄ octamer). **c**, AKRtyl bound 2 NADP(H)s in the same tetramer (R_{4-N2}T₄ octamer). **d**, AKRtyl bound 4 NADP(H)s in the same tetramer (R_{4-N4}T₄ octamer). **e**, AKRtyl bound 1 NADP(H) each in different tetramers (R_{4-N1}R_{4-N1} octamer). **f**, AKRtyl bound 8 NADP(H)s in different tetramers (R_{4-N4}R_{4-N4} octamer).

The model in Fig 2e does not represent the MWC model, since no equilibrium of ligand-free protein states is shown.

Thanks to the reviewers for pointing out the inappropriateness of our model diagrams and we corrected it as shown in Figure R14. The model diagram illustrates the allosteric pattern within the octamer with the tetramer as the allosteric unit. T conformation is favored at low NADP(H) concentration; the high NADP(H)-binding affinity for R conformation promotes a cooperative compulsory concerted conformational change driven by NADP(H) binding and involving all subunits within tetramers, displacing the equilibrium toward the R conformation. It should be noted that we have not shown the state of the T-conformation subunit bound to NADP(H) because we are not sure if the T-conformation can bind NADP(H) with low affinity, since the structural features show that it is not suitable for cofactor binding.

Thanks for the reviewer's reminder. We have made modifications to the MWC model diagram into the manuscript as shown in Figure 2d.

Figure R14. MWC allosteric model for the octameric AKRtyl. The allosteric oligomeric basic unit is tetramer and the tetramer can populate only two conformational states in equilibrium: all subunits in R conformation or all subunits in T conformation. The upper square brackets indicate the conformational equilibrium of the two tetramers, corresponding to the three states of the octamer: T_4T_4 , R_4T_4 , and R_4R_4 . The square brackets in the second row indicate that one tetramer in the octamer is conformationally stabilized to R_4 upon binding of NADP(H), and the other tetramer is in conformational equilibrium. The T-conformation subunit is shown as a green blob, the R-conformation subunit as a blue blob and the R-conformation subunit with bound NADP(H) as a dark blue blob. Red arrows indicate NADP(H)-driven equilibrium towards the R conformation. Dashed lines and dotted arrows indicate multiple processes.

If the whole octamer is considered, which seems the path of least resistance since that's the oligomeric state in solution, then one can see one tetramer in one conformation (bound to one NADPH molecule) while the other tetramer is in another conformation (without any NADPH). This is not consistent with the MWC model, where all molecules in the oligomer must be in the same state. I cannot find justification for considering the tetramer and not the octamer as the allosteric unit in order to invoke the MWC.

Thanks for the question. The judgments for considering the tetramer as the allosteric unit is based on the following points: (1) The tetramer exhibits ligand-independent T and R conformations, suggesting that the equilibrium between the T and R conformations occurs within the tetramer. (2) The tetramer has structural symmetry in different ligand states (unliganded, partially liganded, fully liganded). And these features fit the MWC model.²⁰

We have added the explanations into the manuscript (line 237-242).

In Extended Data Fig 4 and Supplementary Table 3, k_{obs1} and k_{obs2} are reported from the double-exponential fit. Assuming k_{obs1} is reporting on the cooperative binding of NADPH (k_{obs2} is assumed to be the binding of NADPH to the substrate inhibition site), a replot of k_{obs1} vs [NADPH] concentration could help distinguish between MWC and KNF. I'm surprised the authors did not do that. A hyperbolic decrease in k_{obs} with concentration is diagnostic of MWC (it does not seem to be the case from the table), while a hyperbolic increase is common for KNF, but can also be obtained for MWC under certain conditions, which can be experimentally tested. See, for example, Vogt and Di Cera Conformational Selection or Induced Fit? A Critical Appraisal of the Kinetic Mechanism, *Biochemistry* 2012 for details.

Thanks to the reviewer for the suggestion and question. The work of Austin D. Vogt and Enrico Di Cera gives a more rigorous and critical appraisal of the kinetic mechanism to judge whether it is conformational selection or induced fit. The conformational selection case discussed by the researchers is similar to the simplest

form of the MWC allosteric model, while the induced fit case is similar to the simplest form of the KNF model. They showed that in the induced fit model k_{obs} always increases as [L] increases. However, conformational selection is associated with a rich repertoire of kinetic properties, with k_{obs} decreasing or increasing with [L] depending on the relative magnitude of the rate of ligand dissociation, k_{off} , and the rate of conformational isomerization (unfavorable conformation \rightarrow favorable conformation), k_r . When $k_{off} > k_r$, k_{obs} decreases as [L] increases, whereas when $k_{off} < k_r$, k_{obs} increases as [L] increases. Therefore, an increase in k_{obs} with [L] is not unequivocal evidence of induced fit (the simplest form of the KNF).

Based on the two models proposed in the reference, we fitted the k_{obsI} -[NADPH] data and found that both models did not fit well because the fitted values for some parameters are outrageous. We speculated the reasons as (1) the mechanism of AKRtyl may be more complicated than the conformational selection model (the simplest form of the MWC) or the induced fit (the simplest form of the KNF), and (2) the division between k_{obsI} and k_{obsII} in the double-exponential decay equation for fitting fluorescence quenching traces may not be as accurate (Figure R). However, the conformational selection model fitting gave a $K_{d,app} = \sim 0.79 \mu\text{M}$, and the induced fit model fitting gave a $K_{d,app} = \sim 15.01 \mu\text{M}$. The $K_{d,app}$ value obtained from the conformational selection model fit is closer to the $K_d = 0.45 \mu\text{M}$ (Figure 2b in the main text) that observed in the fluorescence quenching experiments in equilibrium solution. (as Figure R15 shown). Therefore, we think that conformational selection model may be more appropriate.

Thanks for the reviewer's suggestion. We have analyzed our data using the two models and integrated into manuscript (line 304-309) and Supplementary Fig. 15.

The conformational selection model:

$$k_{obs} = \left[k_{-r} + k_r + k_{off} + k_{on}[L] - \sqrt{(k_{off} + k_{on}[L] - k_{-r} - k_r)^2 + 4k_{-r}k_{on}[L]} \right] / 2$$

$$K_{d,app} = \frac{k_{off}}{k_{on}} \left(\frac{k_r + k_{-r}}{k_r} \right)$$

The induced fit model:

$$k_{obs} = \left[k_{-r} + k_r + k_{off} + k_{on}[L] - \sqrt{(k_{off} + k_{on}[L] - k_{-r} - k_r)^2 + 4k_r k_{off}} \right] / 2$$

$$K_{d,app} = \frac{k_{off}}{k_{on}} \left(\frac{k_r}{k_r + k_{-r}} \right)$$

Where k_{obs} is the observed rate constant, $[L]$ is the ligand concentration, k_r and k_{-r} refer to the rate of conformational isomerization and backward, respectively, k_{on} and k_{off} refer to the rate of ligand association and dissociation, respectively, $K_{d,app}$ is the apparent dissociation constant and the parameter accessible to experimental measurements of the system at equilibrium.

Figure R15. Conformational selection model (a) and induced fit model (b) fitting for k_{obs} -[NADPH] data.

Lines 341-343: The observation of tylosin binding to form a binary complex with AKRtyl does not rule out an ordered mechanism where NADPH must bind first. The AKRtyl:tylosin binary complex may be a dead-end complex. The kinetic mechanism can only be challenged if this new binary complex is shown to be a part of the reaction conducive to products.

Thanks to the reviewer for pointing out this important issue. In our model, the catalytic mechanism is still an ordered bi-bi reaction, and the catalytic process is still NADPH binding first. We have removed the inappropriate sentence.

Supplementary Fig 15a, Supplementary Fig 16a, Supplementary Fig 16c 4-nitrobenzaldehyde, fit to the Hill eq seems like a stretch. Please show how poor the fit to the MM eq is.

Thank for the suggestion. We have plotted M-M equation and Hill equation fit curves for all kinetic data and labelled the R^2 values for each fit (as shown in

Supplementary Fig 23 and Supplementary Fig 24).

Line 756: What were the [NADPH]? Since NADPH absorbs at 340 nm, how was the expected inner filtered effect on the Trp fluorescence (presumably what is being measured) controlled for or corrected for? The same experiments titrating the same [NADPH] into Trp alone is the usual way to detect this.

We varied the NADPH concentration from 2.5 μM to 100 μM in the NADPH binding measurements as shown in the Extended Data Fig 4.

We appreciate the important suggestion to consider the internal filtering effect of NADPH on tryptophan fluorescence. We therefore re-performed the stopped-flow assays of NADPH binding, and titrated tryptophan with the same NADPH concentration as controls. There are 9 tryptophan residues in AKRtyl, so the tryptophan concentration for our control experiment was 1.8 μM (AKRtyl, 0.2 μM). We found that NADPH does not cause fluorescence quenching of tryptophan as shown in Figure R16. After subtracting controls, the binding curve of fluorescence quenching fitted better with the double-exponential decay equation than with the mono-exponential one (as shown in Figure R17 and Table R6). These results are consistent with our previous observations (Extended Data Fig 4 and Supplementary Table 4).

Figure R16. Representative fluorescence quenching trace of NADPH (50 μM) binding to AKRtyl (0.2 μM) and tryptophan control (1.8 μM).

Figure R17. Representative time course of NADPH (50 μM) binding to AKRtyl (0.2 μM) after subtracting the tryptophan control and fitting with double-exponential decay (red line) and mono-exponential decay equations (blue line).

Table R6. Fit of the real time fluorescent quenching of NADPH to AKRtyl (subtracting the tryptophan control) with two models.

NADPH (μM)	Model	$(\Delta F)_1$ (V)	k_{obs1} NADPH ^c Step2 (s ⁻¹) ^a	$(\Delta F)_2$ (V)	k_{obs1} 2 st NADPH ⁱ Step3 (s ⁻¹) ^b	R^2
100	Double exponential	1.34 \pm 0.03	236.14 \pm 8.05	0.45 \pm 0.04	56.81 \pm 3.03	0.9874
	Single exponential	1.61 \pm 0.04	139.31 \pm 1.55	-	-	0.9760
50	Double exponential	1.62 \pm 0.04	232.75 \pm 7.93	0.58 \pm 0.05	60.48 \pm 3.13	0.9906
	Single exponential	2.00 \pm 0.02	139.28 \pm 1.41	-	-	0.9801
25	Double exponential	1.82 \pm 0.04	196.86 \pm 5.42	0.54 \pm 0.05	53.40 \pm 2.81	0.9929
	Single exponential	2.18 \pm 0.01	129.05 \pm 1.11	-	-	0.9856
10	Double exponential	1.60 \pm 0.06	149.55 \pm 4.53	0.53 \pm 0.06	49.55 \pm 2.91	0.9951
	Single exponential	2.01 \pm 0.01	104.09 \pm 0.70	-	-	0.9912
5	Double exponential	1.06 \pm 0.06	127.12 \pm 6.10	0.58 \pm 0.06	42.48 \pm 2.45	0.9923
	Single exponential	1.53 \pm 0.01	76.88 \pm 0.62	-	-	0.9876
2.5	Double exponential	0.93 \pm 0.01	87.98 \pm 1.68	0.09 \pm 0.01	13.63 \pm 2.93	0.9910
	Single exponential	0.97 \pm 0.01	75.11 \pm 0.60	-	-	0.9875

^a The binding process of the NADPH^c corresponds to the step2 of the three-step model.

^b The binding process of the NADPHⁱ corresponds to the step3 of the three-step model.

Minor comments:

In Fig 1a, an H^+ is also a substrate, which is not depicted, but should be. Furthermore, the reaction is shown as irreversible. Do the authors know this for a fact? In their HPLC experiments, they had molar excess NADPH, so it could lead to full conversion.

Can $NADP^+$ and relomysin be converted to tylosin and NDAPH (plus H^+) by AKRtyl?

Thanks to the reviewer for pointing this, which makes us to rethink the reaction. We then measured the reverse reaction – the conversion of $NADP^+$ and relomysin to tylosin and NDAPH (plus H^+) and it is indeed reversible. However, the whole reaction is reduction dominated, with the reduction rate being more than 2000-fold higher than the oxidation reaction.

We have made modifications to the reaction diagram (Figure 1a).

lines 49 -51: There appears to be a substantial degree of circular logic in the sentence. I suggest the authors rephrase it.

Thanks to the reviewer for the reminder. We have rephrased it. (line 49-51)

Lines 59-61: surely many non-allosterically regulated proteins are thermostabilised by ligand binding, so the converse cannot be evidence for allosteric regulation.

Thanks for pointing this out. Indeed, many non-allosterically regulated proteins are thermostabilized by ligand binding. The reason we made the converse suggestion is based a study reported in reference 33. The authors reported that ligands (inhibitors) binding stabilized AKR1A1 but destabilized AKR1B10, whereas in the presence of $NADP^+$, ligands binding destabilized AKR1A1 but stabilized AKR1B10. Based on these observations, the authors suggest that the effects of ligand binding can be considered as allosteric effects. But further evidence, including structural data, is lacking.

We apologize for not citing the reference properly, we rephrase this sentence in the manuscript (line 59-65).

Also, what are AKR1A1 and AKR1B10? This should be defined/explained for those not versed in this superfamily's nomenclature.

AKR1A1 is a member of subfamily 1A of the AKR superfamily and is widely distributed in all mammalian species.¹³ It can catalyze the reduction of a variety of aromatic and medium-chain aliphatic aldehydes to their corresponding alcohols.²¹ AKR1A1 is widely distributed in various tissues and exerts a wide range of detoxification and detoxification functions and many studies have shown a association with to cancer¹³, schizophrenia²², and osteoporosis²³.

AKR1B10 is a member of subfamily 1B of the AKR superfamily and is highly expressed in epithelial cells of the stomach and intestine.¹⁴ It catalyzes the reduction of aldehydes, some ketones and quinones, and shows high specificity for farnesal, geranylgeranyl, retinal and carbonyls.²⁴ AKR1B10 has been reported to be closely associated with cell carcinogenesis and tumor development by regulating the retinoid acid homeostasis²⁵, and is considered a tumor marker²⁶.

We have defined both enzymes in the manuscript (line 59-65). Thanks for the reviewer's suggestion.

Lines 688-690: add the units for the extinction coefficients.

We are sorry for this mistake. The units for the extinction coefficients are $M^{-1} \cdot cm^{-1} \cdot Da^{-1}$ and we have added them in the methods (line 800-804).

Lines 694-695: how long was the decrease in abs340 monitored for? If more than a few minutes, was evaporation controlled for? This must be stated.

The decrease in NADPH absorbance at 340 nm was monitored for 10 minutes in the enzyme activity assay and for 2 minutes in the kinetic assay. Reaction rates were all calculated as the maximum rate in the pre-reaction period. In all experiments, controls were made to subtract the effects of spontaneous oxidation of NADPH and evaporation of the reaction solution on the absorbance. Thanks to the reviewer for pointing this out and we have included the detail in the method (line 817-825).

In eqs 1 and 2, the initial rate v must be lowercase. All terms must be defined.

Thanks to the reviewer for the reminder. We have changed the initial rate v to lowercase and defined all the terms (line 832-838).

The His-tag does not seem to have been removed from the final proteins. What evidence is there have that the presence of the His-tag did not interfere with catalysis and/or allostery in the enzyme used here? If this hasn't yet been done, it must be shown for at least one of the variants that His-tagged and non-His-tagged enzyme forms have the same activity and allosteric regulation.

We thank the reviewer for this question and suggestion. The His-tag of our protein was designed to be at the N-terminus with the thrombin protease cleavage site in the middle, but when we cut the His-tag with thrombin, we found that we could not cut it off. Analyzing the structure, we speculated that the thrombin protease cleavage site might be too close to the main body of the protein. Therefore, we redesigned the expression vector by adding a 10 amino acid flexible linker (GGGGS)₂ between the thrombin protease cleavage site and the protein. The protein with the added linker could be successfully cleaved by thrombin protease and in this way we cleaved off the His-tag of several proteins, WT, E193A and R257W (Figure R18a). We found that the kinetic behaviour ($K_{0.5}$, k_{cat} and Hill coeff n_H) between His-tagged and non-His-tagged enzyme was highly consistent (Figure R18 b-d and Table R7). And the results of the new kinetic determination are in agreement with those of the original manuscript, i.e., Glu193 and Arg257 are important residues for the binding of tylosin^a at the allosteric site. Mutations in these sites would virtually eliminate the allosteric effect.

Thanks for the reviewer's suggestion. We have integrated the data related to the kinetics of His-tagged and non-His-tagged enzymes into manuscript (line 133-135), Supplementary Fig. 7 and Supplementary Table 3.

Figure R18. Kinetic assays of His-tagged and non-His-tagged forms of AKRtyl WT and its mutants. **a**, SDS-PAGE analysis of the His-tag of AKRtyl and its mutants cleaved by thrombin protease. **b-d**, Kinetic curves of His-tagged (red line) and non-His-tagged (blue line) forms of AKRtyl WT and its mutants, WT(**b**), E193A(**c**), R257W(**d**). These kinetic data were fitted with the Hill equation.

Table R7. Kinetic properties of His-tagged and non-His-tagged enzyme forms of AKRtyl WT and its mutants.

Protein	$K_{0.5}$ (μM)	k_{cat} (s^{-1})	k_{cat}/K_m ($\text{s}^{-1} \mu\text{M}^{-1}$)	Hill coeff n_H ^a	
WT	256.00 ± 20.59	2.08 ± 0.08	8.13×10^{-3}	1.91 ± 0.24	Pos. coop.
WT-cut His ₆ -tag	264.10 ± 19.60	1.90 ± 0.07	7.19×10^{-3}	1.97 ± 0.24	Pos. coop.
E193A	447.50 ± 165.60	1.21 ± 0.16	2.70×10^{-3}	0.92 ± 0.16	Loss of coop.
E193A-cut His ₆ -tag	548.20 ± 111.80	1.21 ± 0.10	2.21×10^{-3}	1.03 ± 0.11	Loss of coop.
R257W	407.00 ± 96.85	1.82 ± 0.18	4.47×10^{-3}	1.15 ± 0.19	Reduced coop.
R257W-cut His ₆ -tag	488.70 ± 73.23	2.12 ± 0.13	4.34×10^{-3}	1.11 ± 0.10	Reduced coop.

^a The Hill coefficient n_H of substrate binding as a measure of cooperativity ($n_H > 1$, positive cooperativity; $n_H = 1$, no cooperativity; $n_H < 1$, negative cooperativity).

In the MD simulations, how were the ionization state of ionizable residues, especially histidines, defined at the outset of simulations?

We thank the reviewer for the question. The simulation is carried out with $\text{pH} \approx 7$. Therefore, the ionization state of residues is decided based on whether the residue is acidic, alkaline, or neutral. Acidic residues, including aspartate (Asp) and glutamate (Glu), are deprotonated and negatively charged with a charge of -1. Among alkaline residues, arginine (Arg) and lysine (Lys) are protonated and positively charged with a charge of +1. Histidine (His) is treated here in its ϵ form. Namely, the N in ϵ position is protonated while the N in δ position is not protonated. The residue is neutral. The rest residues are all not charged.

Figure R19. The ionization state of histidine in the MD simulations.

Supplementary Fig 5b: it's not clear that those data could not fit well to the MM equation.

We have made M-M equation and Hill equation fitting curve presentations for all kinetic data and labelled the R^2 values for each fit (as shown in Supplementary Fig. 6).

Table 1: Many of the reported precision in the values is undercut by the uncertainty. Please make sure precision is reported only to the decimal place uncertainty allows.

Thanks to the reviewer for the reminder. We have confirmed and corrected the precision (as shown in Table 1).

Line 160: replace “close” with “closed”

We are sorry for this mistake. We have corrected it (line 173). Thanks the reviewer for pointing out.

Extended Date Fig 6: the arrow should depart from the bond between the *pro*-R H and C4, not from the H itself. The TS is represented as concerted for hydride and proton transfers. Is that known in this family? If so, cite the reference.

We appreciate the important question. We changed the TS drawing for hydride and proton transfers from a concerted mechanism to a stepwise manner after revisiting literatures. It was reported that hydride and proton transfer could occur in both a concerted mechanism or in a stepwise manner. The specific time gap between the two process dictates, in a large measure, the extent of charge developed on the carbonyl during the transition state and may be the source of differences between the catalytic properties and substrate preferences of different AKRs.²⁷ However, the catalytic mechanism of AKR is generally considered to be a “push-pull” mode. The protonated form of tyrosine (TyrOH²⁺) acts as the general acid, which is facilitated by the protonation state of neighboring tetrad residues, and forms a hydrogen bond with the substrate carbonyl, resulting in carbonyl polarization and accelerating the hydride transfer. Following hydride transfer to the acceptor carbonyl, the carbonyl is protonated by tyrosine via a proton relay that involves tetrad residues to bulk water.²⁸ In this regard, it may be more proper to use the stepwise manner (Figure R2).

Thanks for the reviewer’s suggestion and reminder. We have corrected the arrow problem in the diagram and changed the TS drawing for hydride and proton transfers from a concerted mechanism to a stepwise manner in the manuscript (Extended Date Fig 6).

Figure R20. Proposed AKRtyl-catalyzed reduction mechanism. In complex 1, the

catalytic residue Tyr53 forms a hydrogen bond with the tyrosin carbonyl, resulting in carbonyl polarization and accelerating the hydride transfer of the pro-R hydrogen from the nicotinamide ring of NADPH to the carbonyl carbon of the substrate. The hydrogen bond network provided by His130, Lys85, and Asp48 serves to lower the pKa of Tyr53, facilitating proton transfer. Complex 2 shows a transition state in which after hydride transfer to the acceptor carbonyl, the carbonyl is protonated by tyrosine via a proton relay that involves tetrad residues to bulk water. The reduced carbonyl then dissociates from the acid-base catalyst and a net charge on the tyrosinated anion is stabilized by the hydrogen bonding network (complex 3).

References

1. Bennett BD, Kimball EH, Gao M, Osterhout R, Van Dien SJ, Rabinowitz JD. Absolute metabolite concentrations and implied enzyme active site occupancy in *Escherichia coli*. *Nat. Chem. Biol.* **5**, 593-599 (2009).
2. Goldbeck O, Eck AW, Seibold GM. Real Time Monitoring of NADPH Concentrations in and via the Genetically Encoded Sensor mBFP. *Front. Microbiol.* **9**, 2564 (2018).
3. Tamoi M, Miyazaki T, Fukamizo T, Shigeoka S. The Calvin cycle in *cyanobacteria* is regulated by CP12 via the NAD(H)/NADP(H) ratio under light/dark conditions. *Plant J.* **42**, 504-513 (2005).
4. Huang D, Li SS, Xia ML, Wen JP, Jia XQ. Genome-scale metabolic network guided engineering of for FK506 production improvement. *Microb. Cell Fact.* **12**, 52 (2013).
5. Nagano N, Hutchinson EG, Thornton JM. Barrel structures in proteins: Automatic identification and classification including a sequence analysis of TIM barrels. *Protein Sci.* **8**, 2072-2084 (1999).
6. Banner DW, *et al.* Structure of Chicken Muscle Triose Phosphate Isomerase Determined Crystallographically at 2.5Å Resolution Using Amino-Acid Sequence Data. *Nature* **255**, 609-614 (1975).

7. Nickbarg EB, Davenport RC, Petsko GA, Knowles JR. Triosephosphate Isomerase - Removal of a Putatively Electrophilic Histidine Residue Results in a Subtle Change in Catalytic Mechanism. *Biochemistry* **27**, 5948-5960 (1988).
8. Komives EA, Chang LC, Lolis E, Tilton RF, Petsko GA, Knowles JR. Electrophilic Catalysis in Triosephosphate Isomerase - the Role of Histidine-95. *Biochemistry* **30**, 3011-3019 (1991).
9. Lolis E, Petsko GA. Crystallographic Analysis of the Complex between Triosephosphate Isomerase and 2-Phosphoglycolate at 2.5Å Resolution - Implications for Catalysis. *Biochemistry* **29**, 6619-6625 (1990).
10. Farber GK, Petsko GA. The Evolution of Alpha-Beta-Barrel Enzymes. *Trends Biochem. Sci.* **15**, 228-234 (1990).
11. "Sodium Chloride Injection - FDA prescribing information, side effects and uses". *www.drugs.com*. Archived from the original on 18 January 2017. Retrieved 14 January 2017.
12. Jin Y, Penning TM. Aldo-keto reductases and bioactivation/detoxication. *Annu. Rev. Pharmacol.* **47**, 263-292 (2007).
13. Alzeer S, Ellis EM. The role of aldehyde reductase AKR1A1 in the metabolism of gamma-hydroxybutyrate in 1321N1 human astrocytoma cells. *Chem-Biol. Interact.* **191**, 303-307 (2011).
14. Endo S, Matsunaga T, Nishinaka T. The Role of AKR1B10 in Physiology and Pathophysiology. *Metabolites* **11**, 332 (2021).
15. Kabir A, Honda RP, Kamatari YO, Endo S, Fukuoka M, Kuwata K. Effects of ligand binding on the stability of aldo-keto reductases: Implications for stabilizer or destabilizer chaperones. *Protein Sci.* **25**, 2132-2141 (2016).
16. Guo WL, *et al.* Identification and characterization of a strong constitutive promoter for activating biosynthetic genes and producing natural products in *Streptomyces*. *Microb. Cell Fact.* **22**, 127 (2023).
17. Braig K, *et al.* The Crystal-Structure of the Bacterial Chaperonin Groel at 2.8Å. *Nature* **371**, 578-586 (1994).
18. Yifrach O, Horovitz A. Nested Cooperativity in the Atpase Activity of the

- Oligomeric Chaperonin GroEL. *Biochemistry* **34**, 5303-5308 (1995).
19. Danziger O, Rivenzon-Segal D, Wolf SG, Horovitz A. Conversion of the allosteric transition of GroEL from concerted to sequential by the single mutation Asp-155→Ala. *Proc. Natl Acad. Sci. USA* **100**, 13797-13802 (2003).
 20. Morea V, Angelucci F, Tame JRH, Di Cera E, Bellelli A. Structural Basis of Sequential and Concerted Cooperativity. *Biomolecules* **12**, 1651 (2022).
 21. Penning TM. Human Aldo-Keto Reductases and the Metabolic Activation of Polycyclic Aromatic Hydrocarbons. *Chem. Res. Toxicol.* **27**, 1901-1917 (2014).
 22. Iino K, *et al.* *AKR1A1* Variant Associated With Schizophrenia Causes Exon Skipping, Leading to Loss of Enzymatic Activity. *Front Genet.* **12**, 762999 (2021).
 23. Lai CW, *et al.* A novel osteoporosis model with ascorbic acid deficiency in gene knockout mice. *Oncotarget* **8**, 7357-7369 (2017).
 24. Banerjee S. Aldo Keto Reductases AKR1B1 and AKR1B10 in Cancer: Molecular Mechanisms and Signaling Networks. *Adv. Exp. Med. Biol.* **1347**, 65-82 (2021).
 25. Fukumoto S, *et al.* Overexpression of the aldo-keto reductase family protein AKR1B10 is highly correlated with smokers' non-small cell lung carcinomas. *Clin. Cancer Res.* **11**, 1776-1785 (2005).
 26. Gallego O, *et al.* Structural basis for the high *all-trans*-retinaldehyde reductase activity of the tumor marker AKR1B10. *Proc. Natl Acad. Sci. USA* **104**, 20764-20769 (2007).
 27. Barski OA, Tipparaju SM, Bhatnagar A. The Aldo-Keto Reductase Superfamily and Its Role in Drug Metabolism and Detoxification. *Drug Metab. Rev.* **40**, 553-624 (2008).
 28. Penning TM. The aldo-keto reductases (AKRs): Overview. *Chem-Biol. Interact.* **234**, 236-246 (2015).

REVIEWERS' COMMENTS

Reviewer #3 (Remarks to the Author):

The authors have addressed my concerns. They added numerous experimental results to the manuscript, and I fully support publication.

Response to Reviewers' Comments

We sincerely thank the reviewers for the insightful comments and constructive suggestions to improve the quality of our manuscript. We provide a point-by-point response to all comments from reviewers in **BLUE** font below.

Reviewer #3 (Remarks to the Author):

The authors have addressed my concerns. They added numerous experimental results to the manuscript, and I fully support publication.

We sincerely appreciate reviewer's comments and help on improving the quality of the manuscript.